# Temporal localization of upper extremity bilateral synergistic coordination using wearable accelerometers

Khadija F. Zaidi and Qi Wei

Department of Bioengineering, George Mason University, Fairfax, VA, United States of America

## ABSTRACT

**Background**. The human upper extremity is characterized by inherent motor abundance, allowing a diverse array of tasks with agility and adaptability. Upper extremity functional limitations are a common sequela to Stroke, resulting in pronounced motor and sensory impairments in the contralesional arm. While many therapeutic interventions focus on rehabilitating the weaker arm, it is increasingly evident that it is necessary to consider bimanual coordination and motor control.

**Methods**. Participants were recruited to two groups differing in age (Group 1 ($n = 10$): $23.4 \pm 2.9$ years, Group 2 ($n = 10$): $55.9 \pm 10.6$ years) for an exploratory study on the use of accelerometry to quantify bilateral coordination. Three tasks featuring coordinated reaching were selected to investigate the acceleration of the upper arm, forearm, and hand during activities of daily living (ADLs). Subjects were equipped with acceleration and inclination sensors on each upper arm, each forearm, and each hand. Data was segmented in MATLAB to assess inter-limb and intra-limb coordination. Inter-limb coordination was indicated through dissimilarity indices and temporal locations of congruous movement between upper arm, forearm, or hand segments of the right and left limbs. Intra-limb coordination was likewise assessed between upper arm-forearm, upper arm-hand, and forearm-hand segment pairs of the dominant limb.

**Findings**. Acceleration data revealed task-specific movement features during the three distinct tasks. Groups demonstrated diminished similarity as task complexity increased. Groups differed significantly in the hand segments during the buttoning task, with Group 1 showing no coordination in the hand segments during buttoning, and strong coordination in reaching each button with the upper arm and forearm guiding extension. Group 2's dissimilarity scores and percentages of similarity indicated longer periods of inter-limb coordination, particularly towards movement completion. Group 1's dissimilarity scores and percentages of similarity indicated longer periods of intra-limb coordination, particularly in the coordination of the upper arm and forearm segments.

**Interpretation**. The Expanding Procrustes methodology can be applied to compute objective coordination scores using accessible and highly accurate wearable acceleration sensors. The findings of task duration, angular velocity, and peak roll angle are supported by previous studies finding older individuals to present with slower movements, reduced movement stability, and a reduction of laterality between the limbs. The theory of a shift towards ambidexterity with age is supported by the finding of greater inter-limb coordination in the group of subjects above the age of thirty-five. The group below the age of thirty was found to demonstrate longer periods of intra-limb coordination,

Corresponding author
Khadija F. Zaidi, szaidi8@gmu.edu

with upper arm and forearm coordination emerging as a possible explanation for the demonstrated greater stability.

## INTRODUCTION

The human upper extremity is characterized by inherent motor abundance, allowing a diverse array of tasks with agility and adaptability. This abundance is particularly evident during bilateral coordinated movements, such as throwing a basketball or steering a car. Bilateral coordination can be an important indicator of real-world motor performance, particularly in rehabilitation after stroke, Parkinson's, multiple sclerosis, and other brain injuries (*Woytowicz, Whitall & Westlake, 2016*; *Ueyama et al., 2023*).

Upper extremity functional limitations are a common sequela to Stroke, resulting in pronounced motor and sensory impairments in the contralesional arm. While many therapeutic interventions focus on rehabilitating the weaker arm, it is increasingly evident that it is necessary to consider bimanual coordination and motor control (*Kantak, Jax & Wittenberg, 2017*). Figure 1 depicts examples of daily tasks that require symmetric and asymmetric coordination, such as clapping, driving, or eating with utensils. Bilateral coordination is required to perform symmetric tasks, such as movements where homologous muscles are engaged for the arms to move in-phase, anti-phase, or with complex phasing with one another. The upper extremity's ability to engage in asymmetric tasks, which involve inequal force generation, irregular timing, and coordination between non-homologous muscles, further highlights the need to consider bilateral coordination in rehabilitation (*Woytowicz et al., 2020*).

The complex interaction between two corresponding limbs has sometimes been referred to in the literature as a "synergy". The term is loosely defined as a coordinated action, or as a group of variables demonstrating changes and correlations (*Latash, 2021*). Another use of the term 'synergy' has previously been to describe maladaptive compensatory movements which may arise due to muscle stiffness, reduced range of motion, or reduced agility. For the purposes of this article, the term 'synergistic movement' is used to describe an ongoing dynamic stability in the task-dependent movements of the two upper limbs which is matched temporally and/or kinematically (*Shirota et al., 2016*). Inter-limb coordination is used to describe the spatial–temporal relationships between two or more limbs performing a motor task, while intra-limb coordination describes the coordination between limb segments. Such coordination is significant in upper extremity movements, particularly in patients with functional limitations.

Bilateral synergistic movement is also of interest in studies of age-related differences in motor performance. Aging may lead to deficits in movement control or compensatory adaptations due to natural changes in the neuromuscular physiology such as reduced muscle strength or visuomotor adaptation (*Cauraugh & Kang, 2021*; *Kang et al., 2019*;

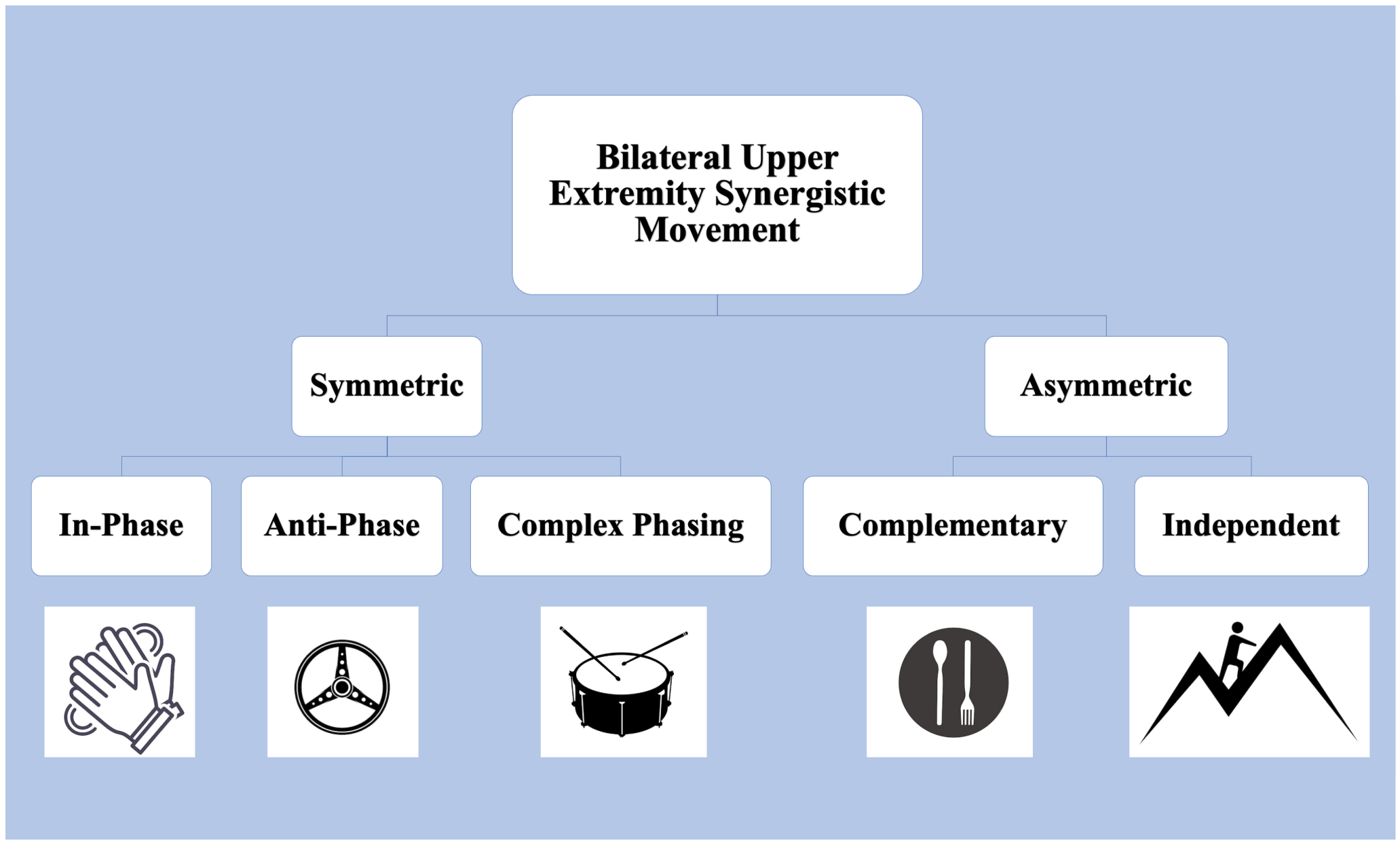

**Figure 1 Upper Extremity tasks requiring bilateral synergistic movement can be symmetric or asymmetric.** In-phase tasks are spatially and temporally matched and equal in force magnitude. Anti-phase tasks alternate spatially symmetric movements with equal forces. Complex phasing tasks are spatially similar but unequal temporally and in force. In asymmetric complementary tasks, forces interact in opposition. Icons from FreePik. Image source credits: Two hands clapping (delvish, License: FreePik Premium); A drum with two drumsticks (terrabismail, License: FreePik Premium); Steering wheel (amin45, License: FreePik Premium); A black plate with white spoon and fork (amin45, License: FreePik Premium).

*Moulton et al., 2022*; *Krehbiel, Kang & Cauraugh, 2017*). As individuals age, they may demonstrate slower and less accurate movements (*Maes et al., 2017*), as well as increased symmetrical performance due to loss of motor lateralization (*Pan et al., 2023*; *Roman-Liu & Tokarski, 2020*; *King et al., 2017*). Recent studies suggest that age-related differences are less evident in the motor system than the underlying neural control processes, playing a significant role in bimanual coordination (*Moulton et al., 2022*; *Heilman, 2019*). Though there is an understanding of the physical impact of aging on motor performance, its impact on bilateral upper limb coordination presents a compelling area of study (*Kang, Ko & Cauraugh, 2022*; *Nouredanesh et al., 2021*; *Roman-Liu & Mockałło, 2020*).

While the idea of coordination is fairly intuitive, it is difficult to define or measure as an objective quantity (*Shirota et al., 2016*). Coordination cannot be quantified by isolating any one kinematic variable, and understandably there is currently no consensus on how to interpret coordination measures. Inter-limb coordination can be subjectively evaluated by analyzing how movement emerges out of task-specific features, the environment, and the individual's motor capability. Traditional methods of assessing the motor function of limb segments include self-assessment questionnaires and clinical assessments (*Larrivée et al., 2021*; *Hulleck et al., 2022*), which cannot evaluate the performance of daily activities and task-specific movement capacity (*Kang, Ko & Cauraugh, 2022*). With developments in

technology, there are now several wearable movement sensors that can objectively and non-invasively evaluate motor performance (*Noorkõiv, Rodgers & Price, 2014*; *Germini et al., 2022*; *Lee, Park & Kim, 2015*). Inertial Measurement Units(IMUs), such as accelerometers, are increasingly available in the form of commercial activity trackers (*Trojaniello et al., 2015*; *Wei et al., 2019*), potentially allowing rehabilitative practices outside of the clinical setting (*De Castro et al., 2021*; *Swanson et al., 2023*; *David et al., 2021*).

In this article, we propose an exploratory study of accelerometers to quantify inter-limb and intra-limb coordination during three tasks requiring bilateral synergistic movement and with increasing complexity. These tasks include reaching to a target, buttoning and unbuttoning a shirt, and reaching up to open a cabinet and retrieve an object. Subjects were placed in two groups differing in age and data was collected from six accelerometers placed on upper arm, forearm, and hand segments of both limbs. Data was then analyzed with a modification of the Reach Severity and Dissimilarity Index (RSDI) methodology described in previous work (*Zaidi & Harris-Love, 2023*). The RSDI is based in Procrustean distance to evaluate age-related differences in coordination by identifying arm trajectories with matching kinematic differences and temporal location.

We expected to see established age-related differences using the RSDI methodology, as well as evident differences in intra-limb and inter-limb coordination between the two groups. We hypothesize that Group 1 will demonstrate greater intra-limb coordination, indicated by lower dissimilarity indices, between the upper arm and forearm segments during movement initiation of symmetric tasks, and towards completion of asymmetric tasks. We hypothesize Group 2 to demonstrate an elongated inter-limb coordination in all tasks, indicated by length of segments with matching temporal locations and congruous trajectories.

The article proceeds as follows: (1) Inclusion criteria for participants and classification into two groups differentiated by age, (2) a description of upper extremity tasks selected to demonstrate symmetric and asymmetric bilateral movements and expected task-specific movement, (3) data segmentation and feature extraction methodologies, (4) resulting quantitative measures of inter-limb and intra-limb coordination, and (5) statistical analyses performed to evaluate differences between age groups and interaction effects.

# METHODS

## Participants

Subjects were recruited from the university student and faculty populations, and screened for any injuries or conditions that may affect upper extremity movement capabilities. This protocol was approved by the George Mason University Institutional Review Board under protocol number (2077208-1). All subjects completed written consent forms.

Subjects that were (1) under 30 years of age, (2) right hand dominant, and (3) able to complete coordinated tasks with both arms including reaching and buttoning a shirt were classified as Group 1. Subjects that were (1) over 35 and under 75 years of age, (2) right hand dominant, and (3) able to complete coordinated tasks with both arms were classified as Group 2. Upon obtaining informed consent, participants underwent a subjective test to confirm their ability to reach forward with both arms.

**Table 1 Subjects were classified to be in Group 1 if under 30 years of age, and in Group 2 if between the ages of 35 and 75.** Subject arm lengths ($p = 0.23$) and eye level height ($p = 0.10$) did not differ significantly between groups.

| Subject | Age (years)* | Gender | Arm length (cm) | Eye height (cm) |
|---|---|---|---|---|
| G1.01 | 24 | M | 67 | 146 |
| G1.02 | 20 | M | 78 | 154 |
| G1.03 | 24 | M | 75 | 142 |
| G1.04 | 29 | M | 73 | 151 |
| G1.05 | 27 | M | 71 | 163 |
| G1.06 | 24 | M | 77 | 154 |
| G1.07 | 23 | F | 75 | 148 |
| G1.08 | 21 | F | 69 | 140 |
| G1.09 | 20 | F | 72 | 140 |
| G1.10 | 22 | F | 75 | 145 |
| Mean | 23.4 | | 73.2 | 148.3 |
| SD | 2.9 | | 3.5 | 7.3 |
| G2.01 | 63 | F | 70 | 137 |
| G2.02 | 57 | F | 61 | 124 |
| G2.03 | 58 | M | 76 | 138 |
| G2.04 | 56 | F | 71 | 147 |
| G2.05 | 57 | F | 69 | 127 |
| G2.06 | 71 | M | 83 | 154 |
| G2.07 | 61 | M | 69 | 143 |
| G2.08 | 61 | M | 72 | 143 |
| G2.09 | 39 | M | 73 | 162 |
| G2.10 | 36 | M | 72 | 150 |
| Mean | 55.9 | | 71.6 | 142.5 |
| SD | 10.6 | | 5.6 | 11.7 |

The sample size was limited to 10 subjects in each test group by the practical considerations of budget and availability of subjects. Subjects who had significant signal loss to their data due to software error were not included. Subjects with signal loss were not asked to repeat the protocol, to avoid the possibility of learning effects or benefits of practice.

The complete group demographics are contained in Table 1. A two-sample $t$-test with equal variance and a one-tailed distribution was performed to evaluate differences between the two groups. Groups differed significantly only in age (Group 1: 23.4 ± 2.9 years, Group 2: 55.9 ± 10.6 years, $p = 1.3E{-8}$), and did not differ in arm length (Group 1: 73.2 ± 3.5 cm, Group 2: 71.6 ± 5.6 cm, $p = 0.23$) or eye height level (Group 1: 148.3 ± 7.3 cm, Group 2: 142.5 ± 11.7 cm, $p = 0.10$).

## Reaching task selection criteria

Three tasks featuring coordinated reaching were selected to investigate the acceleration of the upper arm, forearm, and hand during Activities of Daily Living (ADLs). Tasks were selected to increase in complexity of bilateral coordination, as well as to indicate

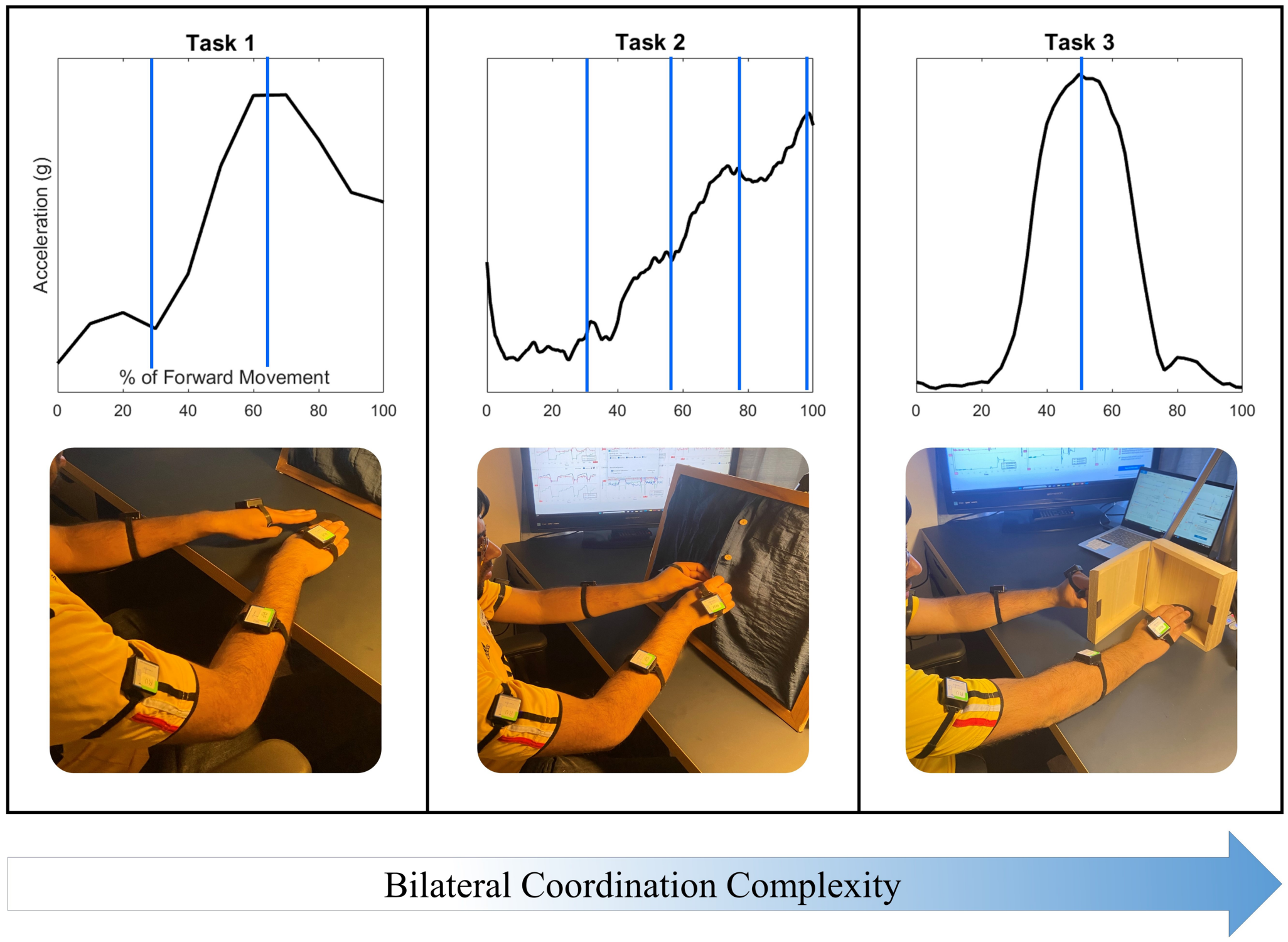

**Figure 2 Three tasks were selected to demonstrate coordination with increasing task and movement complexity.** Task 1: Reaching to a target features symmetry between all limb segments during movement initiation, forward extension, and target precision. Task 2: The buttoning/unbuttoning tasks feature symmetry between the upper arm segments but asymmetry between the hand segments as each of four buttons is accessed and buttoned. Task 3: Finally, the cabinet task features complete asymmetry between all limb segments.

characteristic velocity peaks during the task for the purposes of data segmentation, as shown in Fig. 2. The first task selected was a simple reach-to-target movement to demonstrate an in-phase symmetric task. This task is frequently used in rehabilitative practices for its recruitment of the shoulder, elbow, and wrist joints, and demonstrates two characteristic peaks in the movement of each arm (*American Stroke Association, 2019*), with the first peak indicating the initial acceleration of the upper arm, and the second peak indicating the acceleration of the forearm forward (*Elliott et al., 2020*).

The second task selected featured unbuttoning a vertical column of four buttons to demonstrate a complex-phasing symmetric movement, and then buttoning the same column of buttons to demonstrate a complementary asymmetric movement. Complex-phasing tasks are often difficult to measure with optical position sensors (*Turk et al., 2023*), and are of interest when evaluating measurement units for the ability to capture a task closely resembling the real-life context of subjects with increasing difficulty and variability (*Hochstenbach-Waelen & Seelen, 2012*). This task demonstrates four characteristic peaks in both arms, synchronized to the time-location of the hands approaching each of the four buttons.

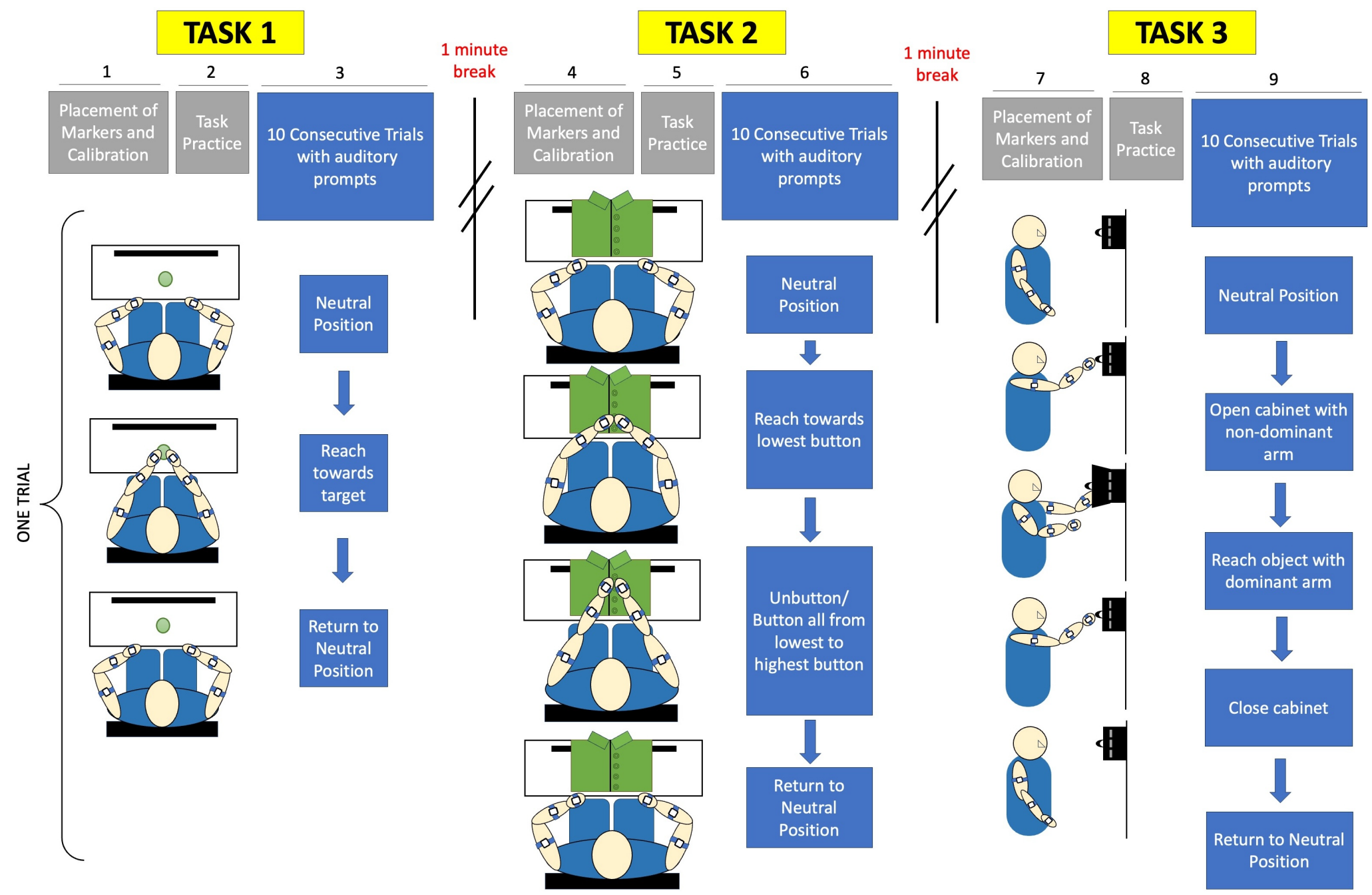

**Figure 3 Experimental protocol included calibration and placement of markers before each task, followed by a single practice trial.** Subjects were given verbal prompts throughout sets of 10 trials for each of three tasks. Task 1 is a reach-to-target task, Task 2 is a buttoning task, and Task 3 is an object retrieval task.

One of the criteria for reaching task selection was the ability to represent movements that are common in activities of daily living that are not represented on clinical assessments. Simulated activities are inadequate representations of upper extremity movements, with functional movements providing a more objective representation of motor ability (*Taylor et al., 2018*). The third task selected required reaching up to open a cabinet placed at eye level with the non-dominant hand, and touching a target placed inside with the dominant hand. This final task was selected to demonstrate an independent asymmetric stabilization movement. Asymmetrical stabilization tasks are not included in any clinical assessments due to the difficulty of measuring such tasks and the increased complexity of the task. The cabinet task also represents a lack of bilateral coordination due to distinct demands on each upper limb; the non-dominant arm demonstrates two acceleration peaks while the dominant arm demonstrates a single acceleration peak.

## Experimental protocol

In order to collect acceleration and inclination data non-invasively, each arm was equipped with three BWT901CL 9-axis gyroscope, inclinometer, accelerometer sensors manufactured by Wit-Motion (*WitMotion Shenzhen Co., 2014*). The experimental protocol is illustrated in Fig. 3.

The subjects were instructed to turn off all electronic devices to minimize distractions. For the reaching task, the subjects were instructed to sit comfortably on a chair with a target

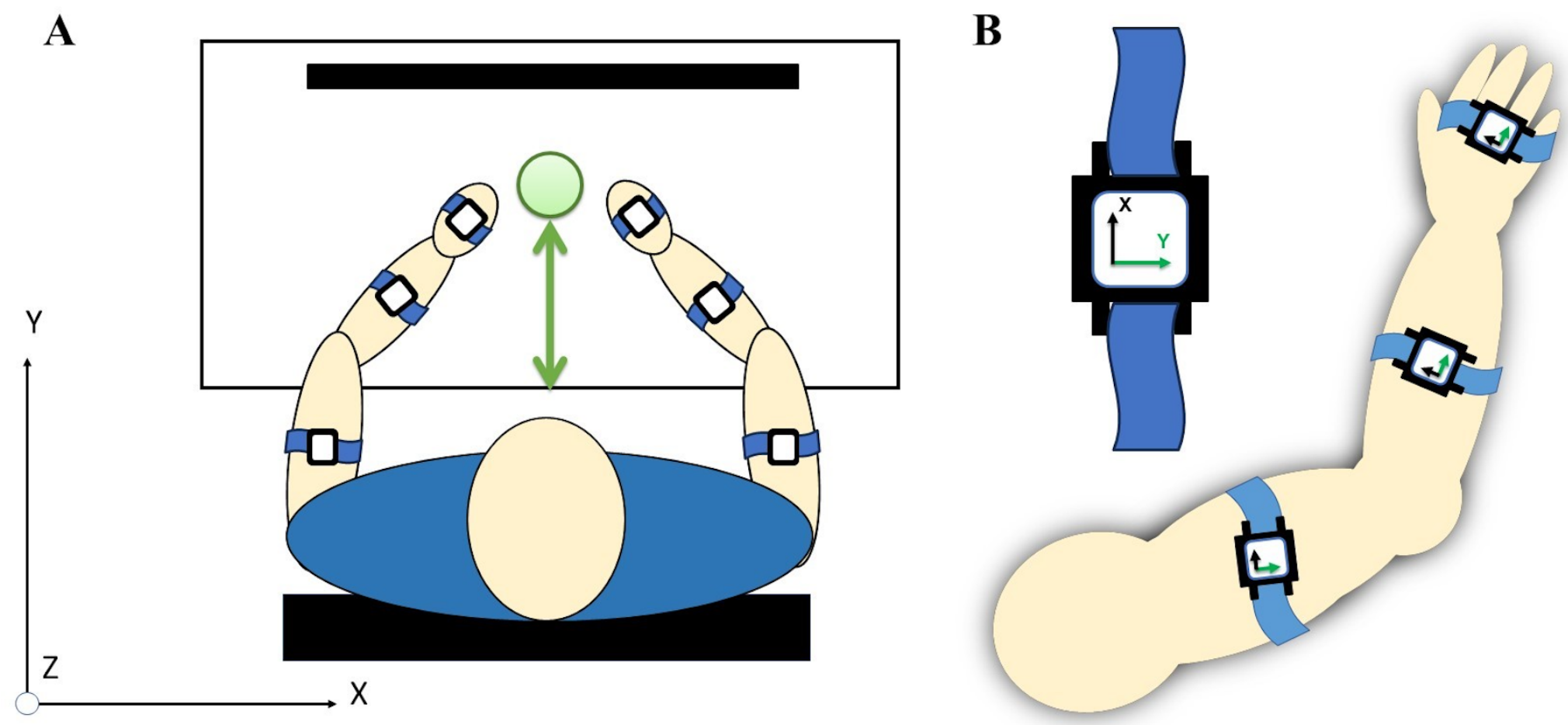

**Figure 4** **(A) A total of six acceleration sensors were worn on upper arm, forearm, and hand segments with velcro straps. (B) Each sensor's local coordinates were aligned for the positive *y*-axis to correspond with the forward direction of a reaching movement.**

positioned centrally within arm's length. They were asked to adjust their seat to ensure a comfortable distance to the workspace, without requiring shoulder extension to reach the target. A calibration procedure was performed following the placement of acceleration sensors, as shown in Fig. 4. Subjects responded to verbal prompts to begin each of ten trials of reaching towards the target from a neutral position at their knees, and were instructed to do so at a comfortable, natural pace.

Subjects were given a 1-minute break between the first and second tasks, providing a break duration that is similar to the required time to complete a task in order to reduce the influence of fatigue (*Dupuis et al., 2022*). Following a break, the subjects proceeded to the second task. During the buttoning task, the subjects were provided with a buttoned shirt affixed to a panel. The subjects were seated with hands placed at a neutral position at the edge of the workspace while the panel with two fabric sides, each containing a series of four buttons, was placed in front of them within 80% of arm's reach. Prior to data collection, a calibration procedure was conducted. The subjects alternated buttoning and unbuttoning the fabric sides, starting from the lowest button and progressing to the highest button, for ten trials. Subjects were once again verbally prompted and instructed to complete the task at a natural and comfortable pace rather than prioritizing speed.

After an approximately 5-minute break, the subjects proceeded to the third task. For the cabinet task, the subjects were asked to place a small magnetic cabinet on a whiteboard so the cabinet is at eye-level. A calibration procedure was performed before data collection. Each of ten trials began with the subject's arms positioned at their sides. They then extended the non-dominant arm to open the cabinet while using the dominant arm to touch a small object inside. To complete each trial, the arms had to return to their initial position. The movement was performed at a natural and comfortable pace.

## Data segmentation and analysis

Data was collected from all three tasks at a frequency of 10 Hz. The sensors had a resolution of 0.005 g for acceleration, 0.61°/s for angular velocity, and 0.05° for angle. Data from each task was segmented with respect to task features and the upper arm. For task 1, a complete reach was defined by a local maxima, preceded and followed by the minimum value of acceleration, corresponding to arms progressing from and returning to the neutral position. For task 2, a complete buttoning or unbuttoning movement was defined by the segment of data between a local minima and the next local maxima, corresponding to hands at the neutral position and progressing to the highest button. Finally, for task 3, a complete reaching movement was defined as the segments between two consecutive local minima, corresponding to the neutral position of arms by sides. Any signal loss was addressed using linear interpolation, and all data was segmented using custom written MATLAB scripts. Length of each task trial, the range of angular velocity, and the angles achieved by upper arms, forearms, and hand segments were calculated from the averaged representative curves for each subject.

### *Inter-limb coordination*

We utilized a novel methodology, referred to henceforth as Expanding Procrustes, to identify segments of the mean representative curve for each task for each subject that were spatially and temporally matched. Using Procrustean distance and dissimilarity indices, the left upper arm movement was evaluated for sub-segments that demonstrated the same trajectories as the right limb during the same phase of movement. Traditional Generalized Procrustes analysis is a widely accepted technique for shape comparison and alignment, validated for use in gait analysis (*Rida, Almaadeed & Almaadeed, 2019*; *Anwary, Yu & Vassallo, 2019*) and upper extremity movement studies (*Passos et al., 2023*; *Saenen, Orban de Xivry & Verheyden, 2022*). This methodology is a further modification of the Reach Severity and Dissimilarity Index, developed previously to quantify impairment severity and dissimilarity in cohorts of persons with stroke.

The Procrustes analysis compares each element of a curve Y to each element of a reference curve X such that a conformed curve Z is produced minimizes the sum of squared Euclidean distances between curves X and Y by translating, rotating/reflecting and scaling the quantities in Y (*Gower, 2004*). Given a two-dimensional set Y of k points, the translation component c is extracted by translating the means to the origin, such that

$$(x_1, y_1) \rightarrow (x_1 - \overline{x}, y_1 - \overline{y})$$

where $\overline{x}$ is the sum of all x-components divided by k, and $\overline{y}$ is likewise the mean of the y-components of all points. The scaling component is extracted by determining the scale that minimizes the root mean square distance (RMSD) of the points from the translated origin to 1.

RMSD:

$$s = \sqrt{\frac{(x_1 - \bar{x})^2 + (y_1 - \bar{y})^2 + \cdots}{k}}.$$

The rotation/reflection matrix T required to conform the curve Y to the reference curve X is found by identifying angle $\theta$ such that:

$$(u_1, v_1) = (\cos\theta\, w_1 - \sin\theta\, z_1, \sin\theta\, w_1 + \cos\theta\, z_1)$$

where $(u_1, v_1)$ is the rotated data point. The conformed curve Z is then written as,

$$Z = b * Y * T + c.$$

where b is the scaling component, T is the rotation/reflection matrix, and c is the translation component. Finally, the Procrustes dissimilarity between curve X and Z is calculated as the square root of all sum distances between corresponding points in the aligned curves.

$$D = \sqrt{\sum_{i=1}^{n}\sum_{j=1}^{k}(a_{ij} - b_{ij})^2}$$

The dissimilarity index D is standardized to produce a value between 0 (complete congruence) and 1 (complete dissimilarity).

Data was processed to produce six three-dimensional acceleration-time curves for each task for the right and left upper arms, right and left forearms, and right and left hands by averaging the ten trials of each movement. The expanding Procrustes methodology was applied to produce dissimilarity scores between the upper arm pair, forearm pair and hand segment pairs during each task. A sliding window was sized to 10% of the overall movement curve and used to systematically traverse the time series data for the right and left limb movements, calculating the Procrustean distance for each sub-segment enclosed by the sliding window. This allowed for identification of those sub-segments between paired right and left curves that yield the lowest Procrustean distance, signifying the highest area of spatial congruence.

Following the identification of the most congruent sub-segments, segments were expanded forward point-by-point to find the maximum time index where the dissimilarity index remains below a threshold representing a lower to moderate level of dissimilarity. The threshold was selected with respect to a data set of individuals with 0.15 being the average dissimilarity index between stages of a simple reaching task. After identifying the maximum time index, segments were expanded backwards point-by-point to identify the minimum time index where the dissimilarity of the segments remains below the threshold. This method resulted in the temporal indices that mark the most congruent trajectories between right and left limb segments.

### Intra-limb coordination

Traditional Procrustes analysis was used to compute an initial dissimilarity reference score within all limb segment pairs and between the dominant and non-dominant limbs. Using expanding Procrustes, intra-limb coordination was evaluated between the three joints pairs for the dominant arm during the two symmetric tasks, and in both dominant and non-dominant arms during the asymmetric third task. Dissimilarity scores were produced between the upper arm and forearm pair, upper arm and hand pair, and the forearm and hand pair. Intra-joint coordination was considered to not be evident in segments

that matched spatial trajectories but did not match temporally to task specific stages of movement.

### Statistical analysis

All statistical analyses were conducted in MATLAB with a statistical significance level of 0.05 (5%). When interpreting the results, $p$-values less than 0.05 were considered statistically significant and suggested the rejection of the null hypothesis. For the purposes of consistency in this article, all statistical analyses were performed using independent t-tests and N-way ANOVA.

Metrics between groups of angular velocity, peak roll angle, and percentage of similarity in inter- and intra-limb coordination were assessed with independent t-tests with a one-tailed distribution and equal variance. A one-tailed distribution was used due to the established expectation that changes in speed and smoothness of movement will occur in one direction with age. This was done to provide support for the expectation that accelerometry would show the same differences between age groups as observed in prior literature.

After coordination scores were calculated from the dissimilarity scores and percentages of similarity for each subject, they were logarithmically normalized. A two-way ANOVA was conducted to identify interaction effects and whether observed differences were due to the increasing task complexity or due to the age differences in the two groups. This was to support our hypotheses that differences between the coordination of the two groups would primarily be due to age differences, and that the differences would appear as increased percentages of similarity and time when the coordination occurs. Additionally, Quantile-Quantile plots were created from the residuals generated by the ANOVA to assess if the data plausibly followed a normal distribution.

## RESULTS

All task-specific differences were evident in the positive $y$-axis in the primary direction of movement during the reaching required in all tasks. For the purposes of data segmentation and feature extraction, the acceleration curve of the right upper arm in the positive y-direction was used. For the purposes of all figures and results, acceleration, angular velocity, and peak angle are calculated and depicted with respect to the positive $y$-axis.

Figure 5 shows the upper arm, forearm, and hand acceleration data during ten trials of Task 1 for a representative subject from each group. All red lines in figure indicate the right dominant limb, while blue lines show the movement of the left non-dominant limb. During the symmetric reach-to-target task, right and left limbs appeared to have congruent trajectories and temporal locations regardless of age group.

Figure 6 shows the upper arm, forearm, and hand acceleration data during ten trials of Task 2 for the same representative subjects as in Fig. 5 from each group. Each local maxima peak evident during the upper arms trajectory indicates the completion of buttoning/unbuttoning. In the Group 1 subject example, the upper arms and forearms appear congruent in trajectory, while the complex phasing of the task is evident in the hand trajectories. During the buttoning task, the left hand and right hand completed

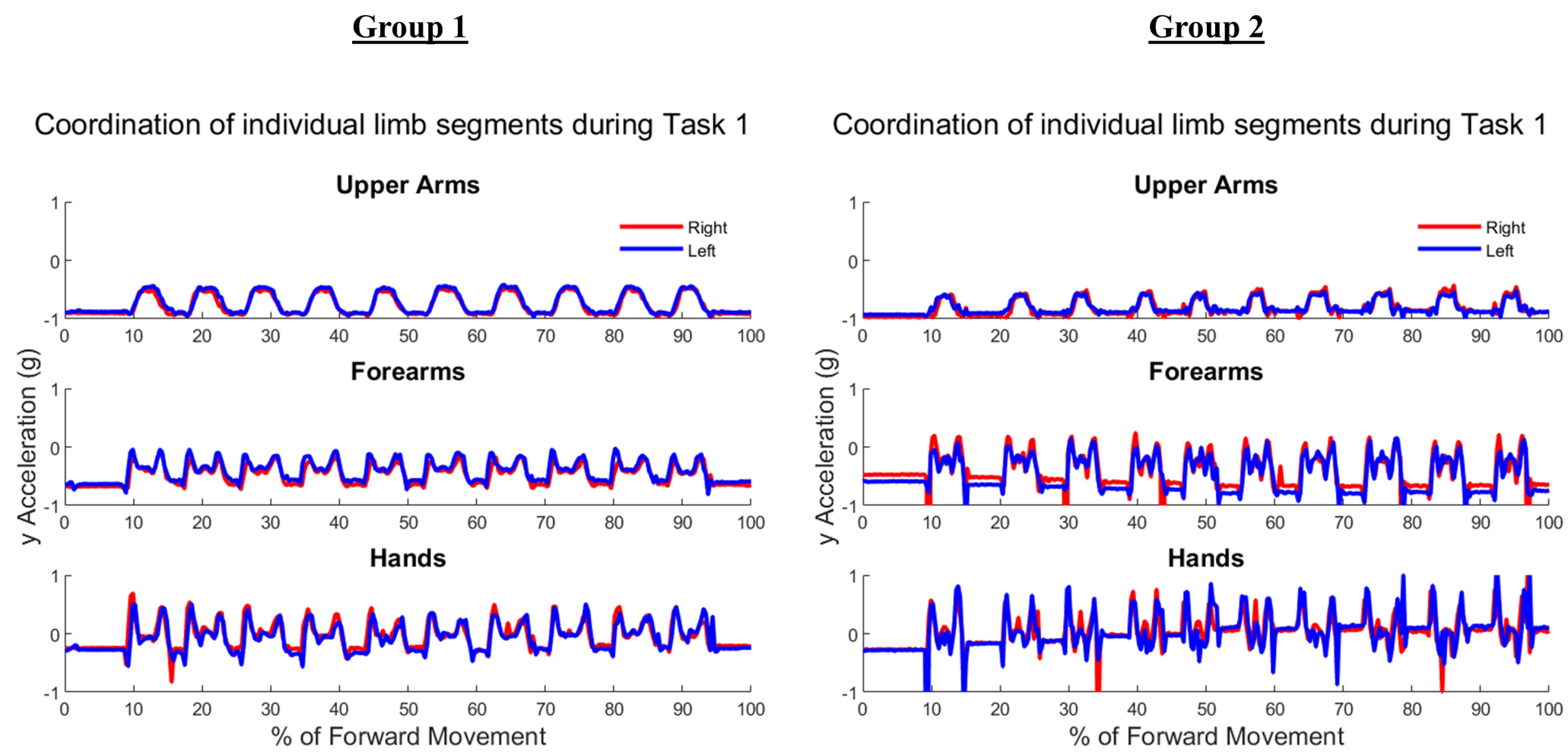

**Figure 5** **Example forward acceleration data in the positive y-direction during Task 1.** The reach-to-target task is a symmetric in-phase task, with right and left limbs coordinated in subject examples from Group 1 and Group 2.

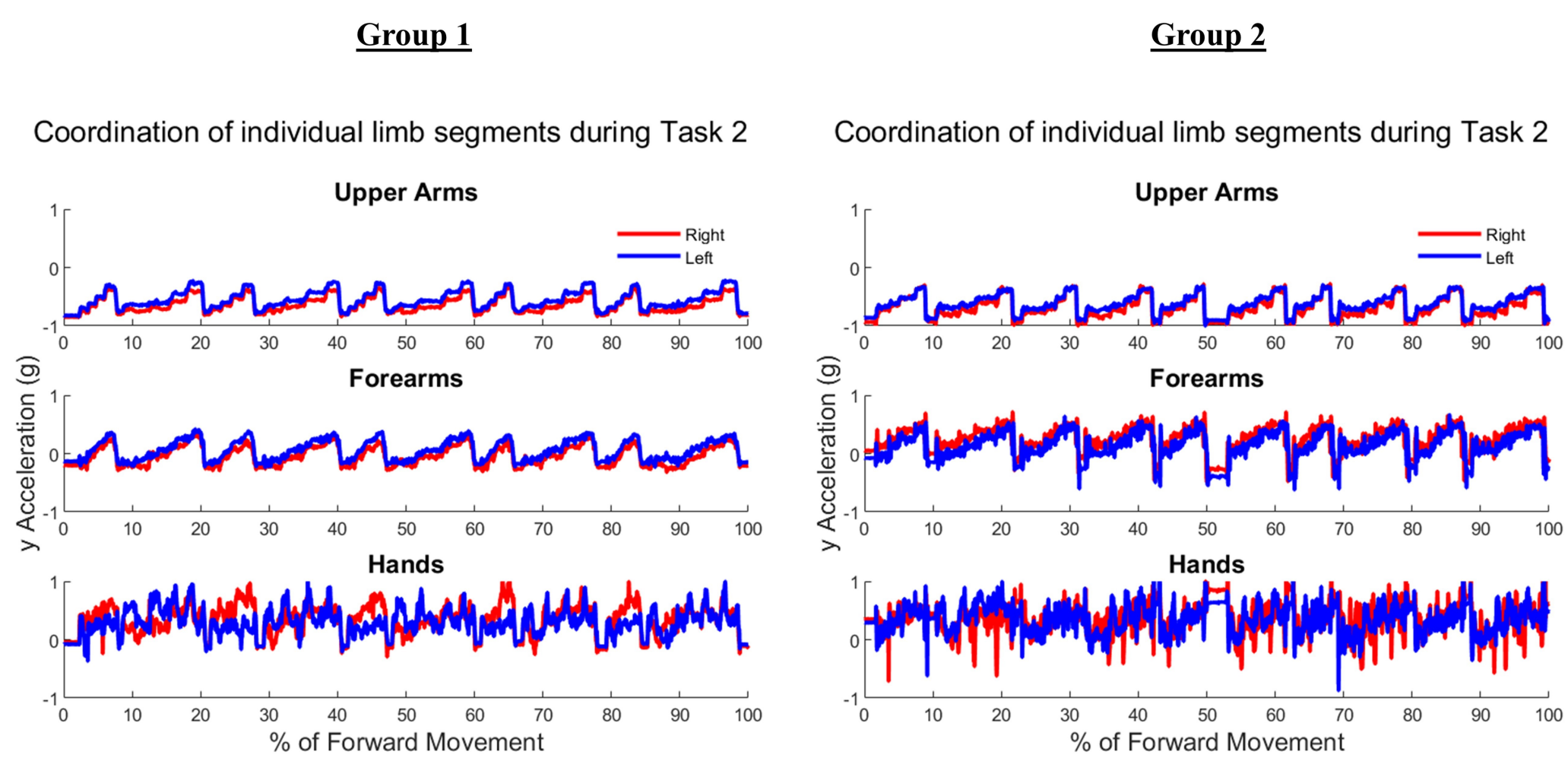

**Figure 6** **Forward acceleration data during Task 2.** The buttoning/unbuttoning task is a symmetric in-phase task for the upper arm and forearm segments, and asymmetric with complementary phasing for the hand segments. Right and left limbs appear to display bilateral coordination in the upper arm and forearm, but not the hands in subject examples from Group 1 and Group 2.

asymmetric tasks in a complementary fashion. Acceleration graphs showed task-specific features such as progression between each button, as well as differences during buttoning and unbuttoning.

Figure 7 depicts ten trials of Task 3 performed by the same representative subject examples from each group shown in Figs. 5 and 6. Task 3 was an asymmetric independent task, which was evident in graphs of Task 3 where right and left limb did not appear to be synchronized in trajectories. As the right limb was used only to reach into and out of

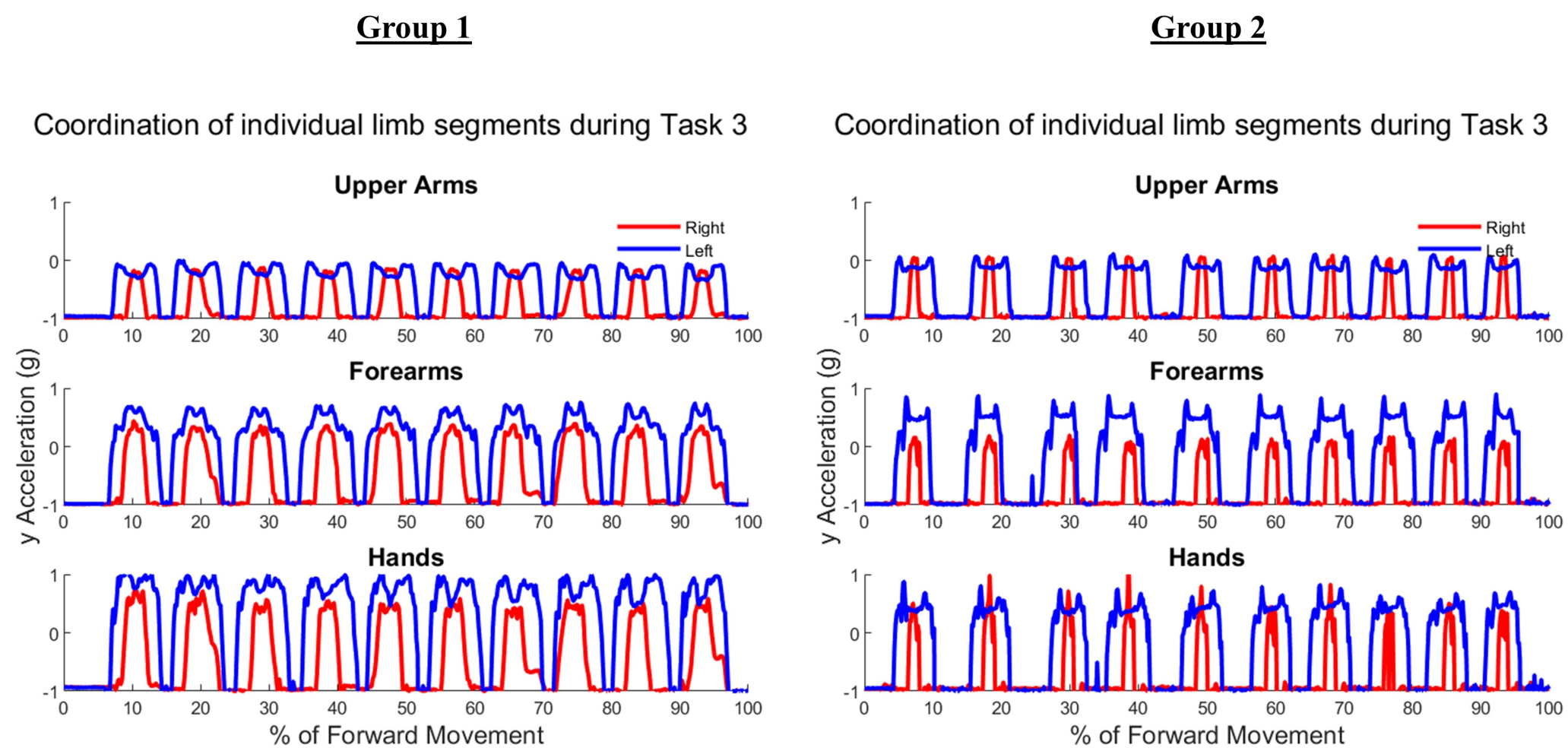

**Figure 7 Forward acceleration data during Task 3.** The cabinet task is an independent asymmetric task, with right and left limbs displaying no obvious coordination in subject examples from Group 1 and Group 2.

the cabinet, it depicts a typical bell pattern as seen in reach-to-target tasks. The left limb generally showed two peaks in all limb segments to depict the initial reach towards the cabinet, swinging the door open and closed, and then returning to the neutral position by an individual's side.

Table 2 shows the time duration of each task for subjects in Group 1 and Group 2. Groups did not differ in completing the buttoning ($p = 0.08$) and cabinet ($p = 0.48$) tasks. Groups did indicate significant differences during the symmetric reach-to-target task ($p = 0.0087$).

## Angular velocity and range of motion

Tables A1 and A2 contain the comprehensive angular velocity values and peak roll angles for all subjects in Group 1 and Group 2. Figure 8 shows the absolute values of the minimum and maximum angular velocities achieved by each limb segment throughout all tasks to evaluate if there are significant differences in angular velocities between groups. The upper arms experienced the least velocities during the forward reaching tasks, compared to the hand segments exhibiting greater velocities for both groups and both limbs. Figure 9 shows the peak roll angle of each limb segment's angular motion throughout all tasks.

Each acceleration sensor collected angle data relative to a local coordinate system centered at the sensor, with the positive $y$-axis extending in front of the subject. Rotations around the $y$-axis, or the roll angle of the arm, showed differences between the groups. Groups had a significantly different minimum peak roll angle of the upper arm segments (RU Minimum Angle Group 1: $6.21 \pm 9.75°/s$, Group 2: $30.61 \pm 24.62°/s$, $p = 0.0046$) and maximum peak roll angle (LU Maximum Angle Group 1: $6.57 \pm 6.54°/s$, Group 2: $23.54 \pm 25.61°/s$, $p = 0.029$).

**Table 2 Group 1 and Group 2 average times to complete one trial of each task.** Groups differed significantly only in Task 1 ($p = 0.01$).

| | **Task 1 Time (s)*** | **Task 2 Time (s)** | **Task 3 Time (s)** |
|---|---|---|---|
| G1.01 | 3.18 | 15.82 | 5.54 |
| G1.02 | 4.00 | 17.45 | 6.63 |
| G1.03 | 2.89 | 18.10 | 6.78 |
| G1.04 | 3.85 | 18.23 | 6.27 |
| G1.05 | 4.34 | 18.04 | 5.99 |
| G1.06 | 2.30 | 15.93 | 5.34 |
| G1.07 | 2.28 | 16.52 | 4.58 |
| G1.08 | 3.89 | 14.78 | 5.15 |
| G1.09 | 2.19 | 11.18 | 5.55 |
| G1.10 | 3.71 | 17.56 | 7.19 |
| Mean | 3.26 | 16.36 | 5.90 |
| STD | 0.80 | 2.15 | 0.82 |
| G2.01 | 9.35 | 17.52 | 4.15 |
| G2.02 | 6.07 | 22.81 | 8.48 |
| G2.03 | 6.09 | 18.95 | 6.92 |
| G2.04 | 4.14 | 18.57 | 4.83 |
| G2.05 | 2.57 | 12.47 | 4.44 |
| G2.06 | 4.74 | 17.52 | 5.24 |
| G2.07 | 4.60 | 14.52 | 5.39 |
| G2.08 | 4.60 | 20.35 | 5.27 |
| G2.09 | 3.72 | 17.84 | 7.41 |
| G2.10 | 3.65 | 19.56 | 7.10 |
| Mean | 4.95 | 18.01 | 5.92 |
| STD | 1.88 | 2.89 | 1.45 |

## Inter-limb coordination

Table A3 shows traditional Procrustes dissimilarity indices for inter-limb comparisons. Traditional Procrustean distance measures did not show any significant differences between groups. Figure 10 depicts the upper arm, forearm, and hand segments of the right and left limbs. In order to compute inter-limb coordination, the expanding Procrustes method was applied to the right and left upper limbs, the right and left forearms, and the right and left hand segments. In cases where the initial congruent segment was found to have a dissimilarity index larger than the threshold of 0.15, segments were considered to demonstrate a 0 percent of similarity. Both groups demonstrated high percentages of similarity for the first task, particularly in the upper arm (Group 1: 55.2 ± 30.05%, Group 2: 58 ± 38.79%). Groups differed significantly in the percentage of similarity demonstrates by the hand segments during the buttoning task (Group 1: 0% similarity, Group 2: 7.6 ± 12.24, $p = 0.033$). Figure 11 shows a subject exemplar from each group. Graphs depict coordination, indicated by the dashed line, between the upper arms, forearms, and hands only during movement initiation. During the unbuttoning task, groups differed

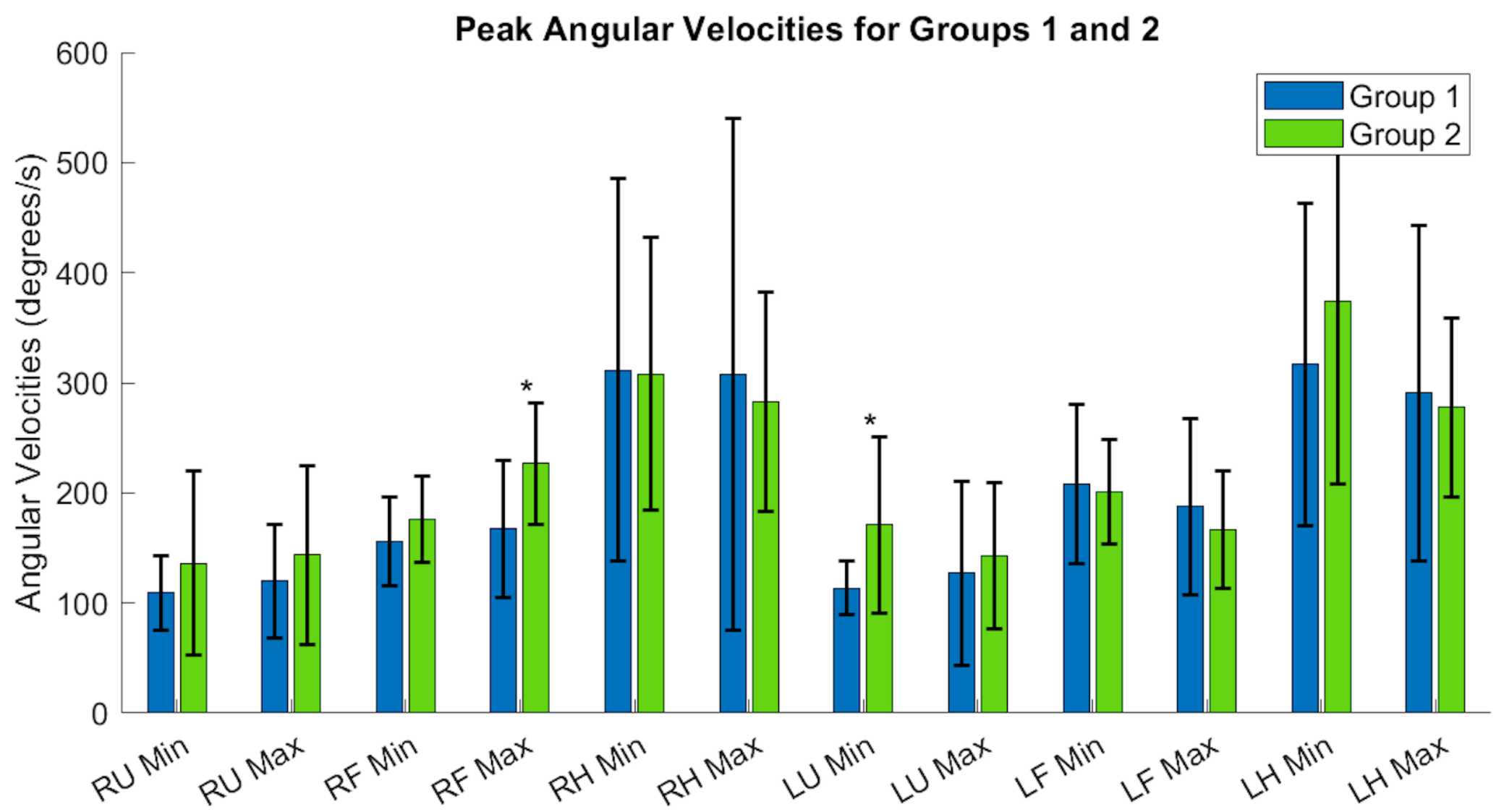

**Figure 8** **Peak angular velocities of each limb segment shows significant differences in the right forearm and left upper arm.** RU, right upper; RF, right forearm; RH, right hand; LU, left upper; LF, left forearm; LH, left hand. Asterisks indicate quantities with significant differences between groups.

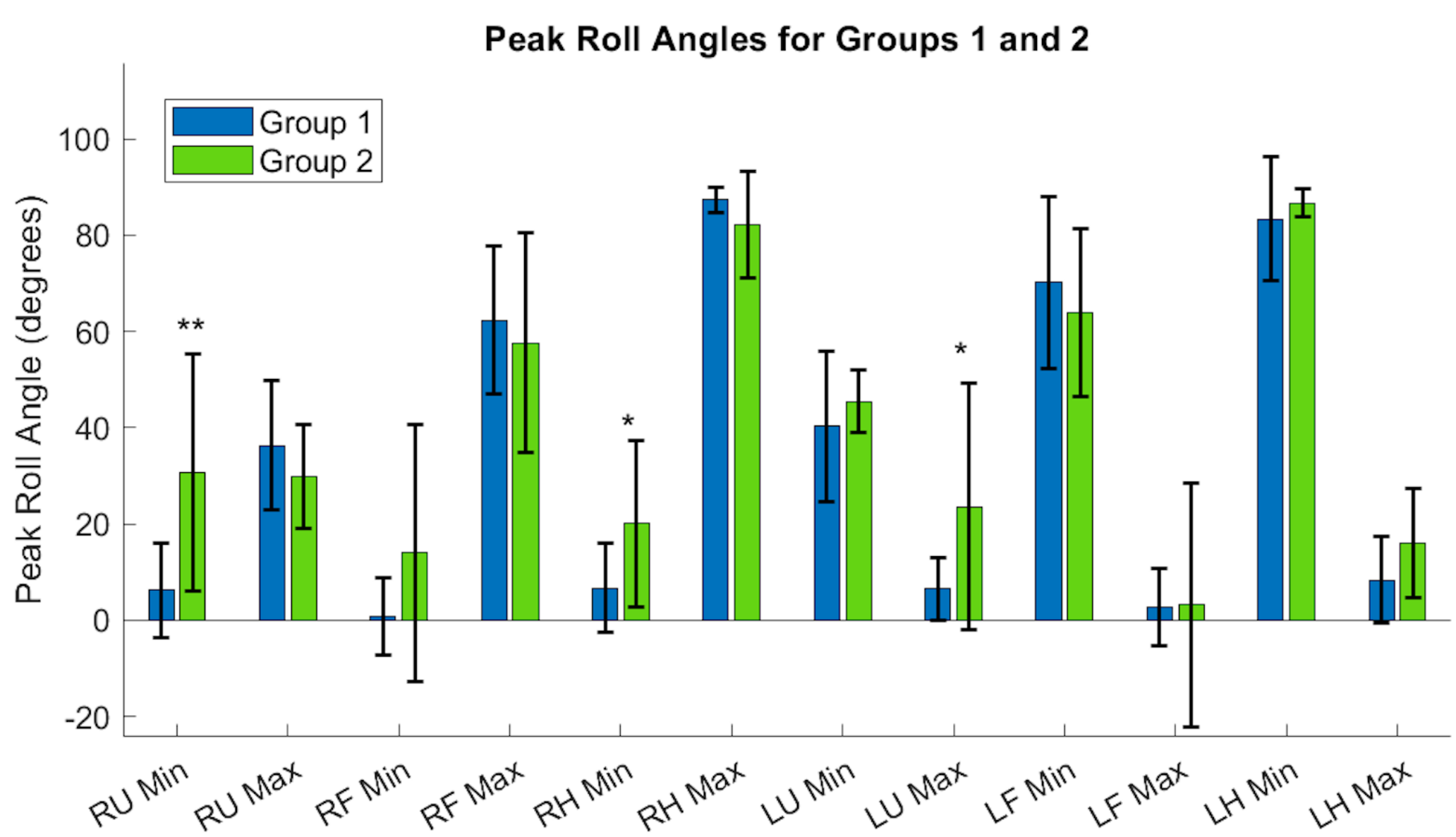

**Figure 9** **Roll angle of the upper arm, or the rotation about the *y*-axis, showed significant differences between Groups, particularly in the right and left upper arms, and the dominant hand.** RU, right upper; RF, right forearm; RH, right hand; LU, left upper; LF, left forearm; LH, left hand. Asterisks indicate quantities with significant differences between groups, with the right upper limb minimum velocity producing a higher significance of differences between groups relative to the right hand and left upper arm segments.

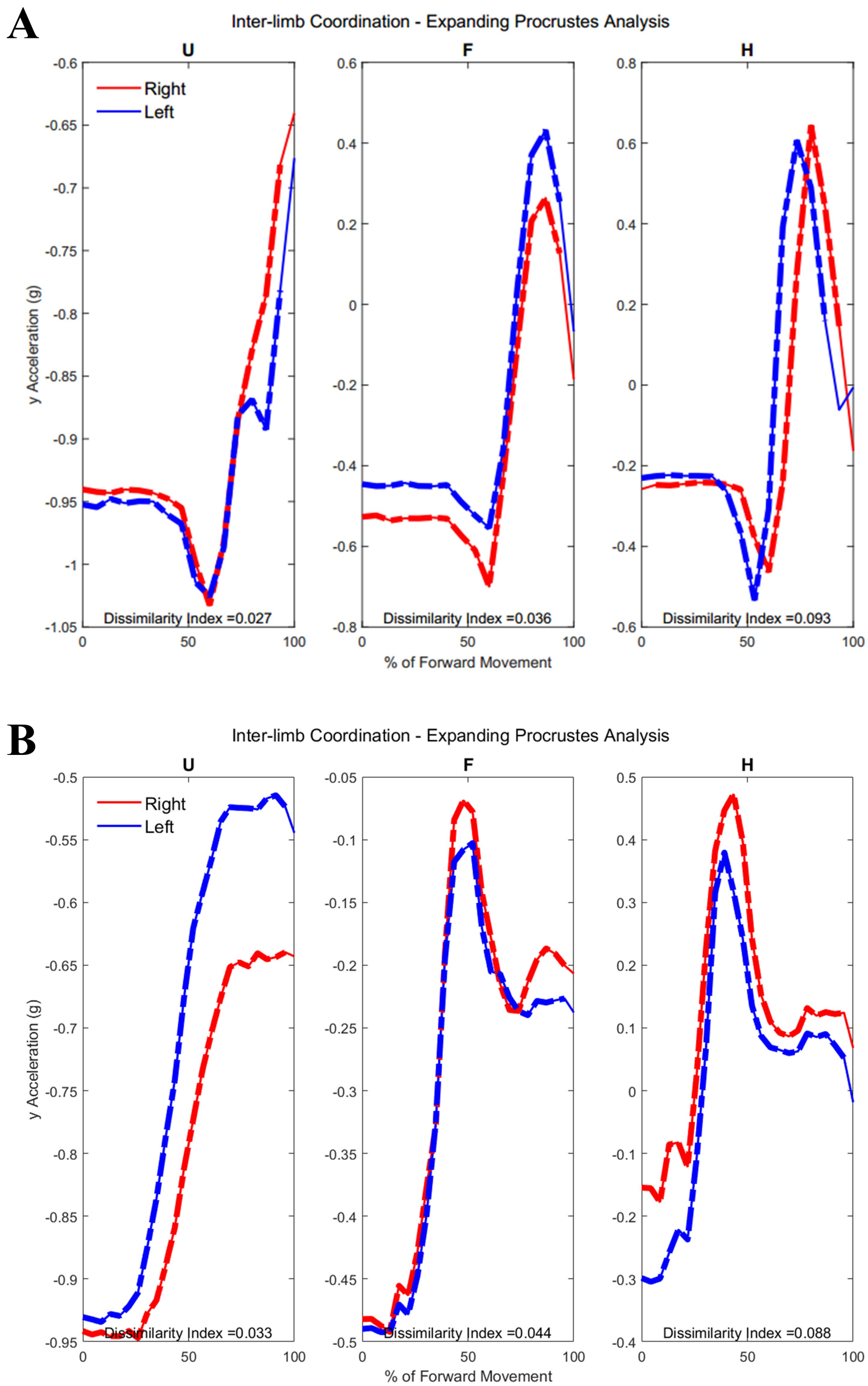

**Figure 10** **(A) Group 1 and (B) Group 2 subject exemplars during Task 1.** U, upper arm; F, forearm; H, hand; with red lines showing the movement of the right limb and blue lines showing the movement of the left limb.

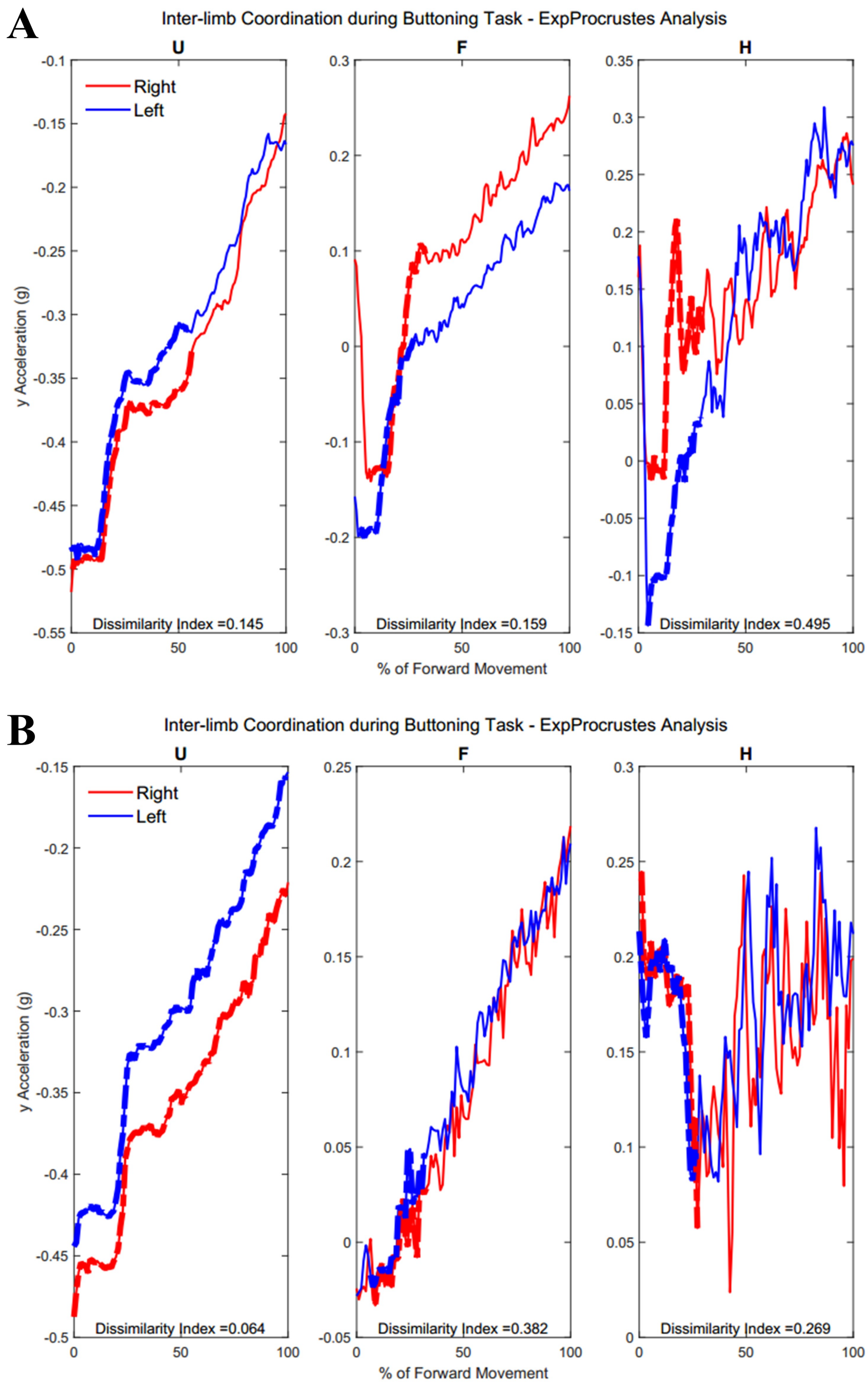

**Figure 11** (A) **Group 1 and B) Group 2 subject exemplars during Task 2.** U, upper arm; F, forearm; H, hand; with red lines showing the movement of the right limb and blue lines showing the movement of the left limb.

significantly again during the movement of the hand segments (Group 1: 0% similarity, Group 2: 7.5 ± 12.08%, $p = 0.033$).

Figure 12 demonstrates the task-specific movements of the right and left limbs during the independent asymmetric third task. In the case of the upper arm, the right limb demonstrated an expected reach-to-target curve, while the left arm produced two peaks during the process of reaching, opening, closing, and retreating from the cabinet placed at eye-level. The right and left limb showed congruent trajectories due to replicating the movement towards the cabinet alternatively between the limbs. Where trajectories did not match temporally, curves resulted in a 0% of coordination similarity. The Group 1 example subject showed no coordination during the independent movements carried out by the right and left limbs. The Group 2 example alternatively showed coordination occurring in all limb segments. Both groups showed diminished coordination in the third task compared to the first two symmetric tasks (Upper arm, Group 1: 17.5 ± 9.65%, Group 2: 16.3 ± 11.11%, $p = 0.40$), (Forearm, Group 1: 29.8 ± 12.27%, Group 2: 12.2 ± 12.18%, $p = 0.0024$), (Hand, Group 1: 19.2 ± 10.40%, Group 2: 16.2 ± 12.56%, $p = 0.28$). During the third task, the two groups differed significantly in the length of similarity between the forearm segments, but not the upper arm and hand segments.

Tables 3 and 4 show dissimilarity indices and percentages of similarity for all subjects in Group 1 and Group 2, respectively. In summary, both groups showed greater lengths of similarity during the first task, and diminished similarity as task complexity increased. Groups differed in the hand segments during the buttoning task, with Group 1 showing no coordination in the hand segments during buttoning, and strong coordination in reaching each button with the upper arm and forearm guiding extension. Groups also differed in the forearm segment coordination during the final cabinet task.

## Intra-limb coordination

Table A4 shows traditional Procrustes dissimilarity indices for intra-limb comparisons. Traditional Procrustean distance measures did not show any significant differences between groups. Figure 13 shows the three segments of the right arm during Task 1. Groups differed significantly in the coordination between forearm and hand segments driving the movement towards the target (Group 1: 52.2 ± 29.87% , Group 2: 23.45 ± 22.53%, $p = 0.013$). Figure 13 shows a subject from each group with more coordination occurring between the hand segments during movement completion.

Tables 5 and 6 show dissimilarity indices between the three joint-pairs (upper arm and forearm, forearm and hand, upper arm and hand) of the dominant limb for Group 1 and Group 2, respectively. No significant differences were found between groups for intra-limb coordination during the buttoning task. Any potential age differences were offset by the complex phasing and symmetry required during buttoning and unbuttoning. Figure 14 shows coordination in the right limb indicated by the dashed lines overlaying the mean acceleration data. Upper arm and forearm segments indicated a greater percentage of similarity between curves during the middle of the movement, when the right hand has reached for the object inside the cabinet and both hands retreat from the cabinet with the

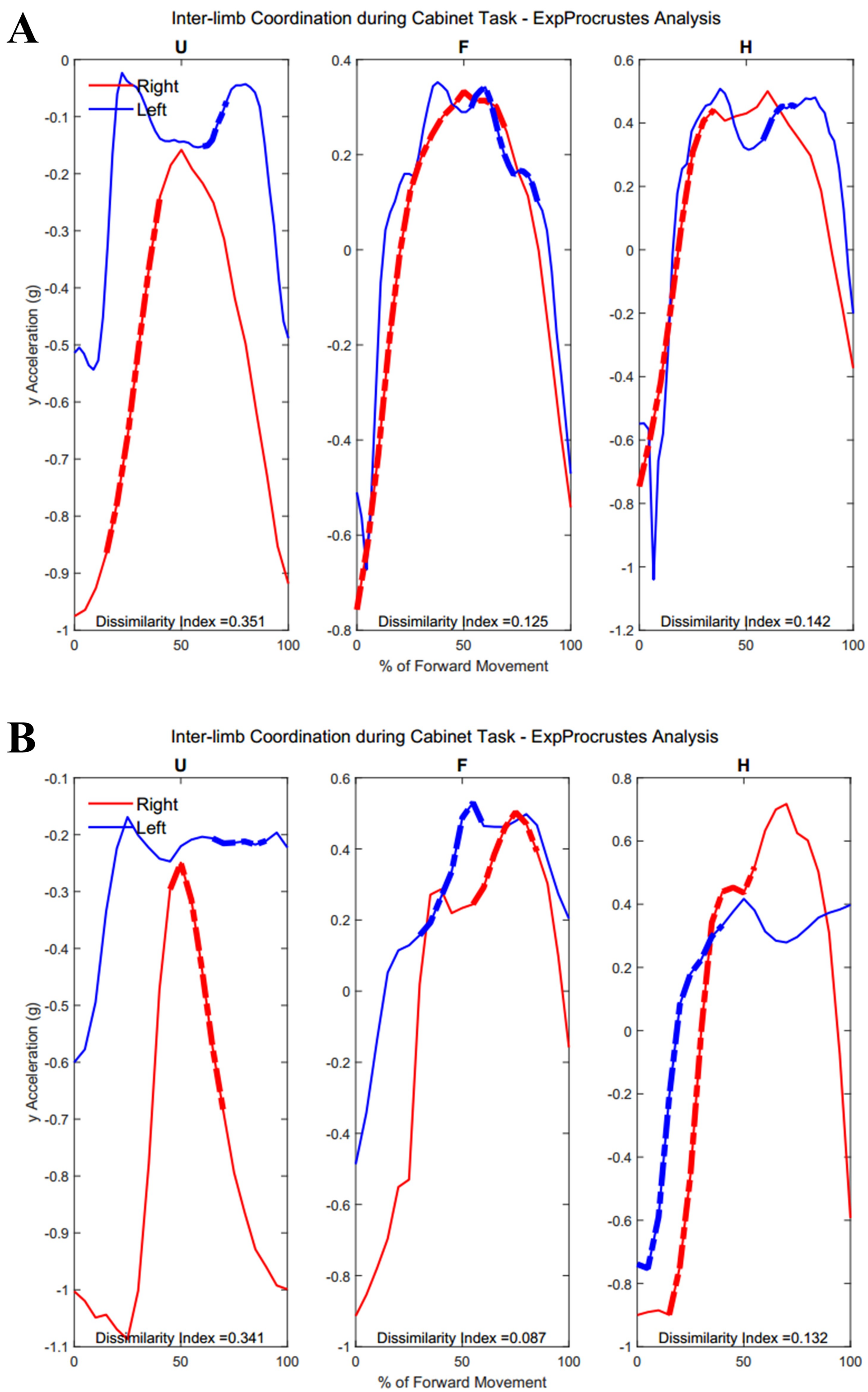

**Figure 12** **(A) Group 1 and (B) Group 2 subject exemplars during Task 3.** U, upper arm; F, forearm; H, hand; with red lines showing the movement of the right limb and blue lines showing the movement of the left limb.

**Table 3 Dissimilarity indices and length of spatially and temporally matched segments in Group 1.** U, upper arms; F, forearms; H, hands; columns 4–6 show the percentage of similarity shown between right and left arms based on minimum and maximum time indices of congruent segments.

| | U EP-DI | F EP-DI | H EP-DI | U % of sim | F % of sim | H % of sim |
|---|---|---|---|---|---|---|
| Task 1 | | | | | | |
| | 0.05 | 0.04 | 0.03 | 80 | 80 | 90 |
| | 0.26 | 0.24 | 0.06 | 33 | 33 | 40 |
| | 0.15 | 0.14 | 0.83 | 45 | 45 | 0 |
| | 0.60 | 0.13 | 0.20 | 0 | 27 | 45 |
| | 0.03 | 0.04 | 0.09 | 93 | 93 | 87 |
| | 0.15 | 0.05 | 0.11 | 38 | 50 | 38 |
| | 0.04 | 0.32 | 0.15 | 88 | 25 | 25 |
| | 0.08 | 0.61 | 0.03 | 83 | 33 | 75 |
| | 0.06 | 0.28 | 0.01 | 38 | 50 | 25 |
| | 0.10 | 0.12 | 0.08 | 54 | 50 | 92 |
| Task 2 (B) | | | | | | |
| | 0.47 | 0.36 | 0.76 | 0 | 25 | 0 |
| | 0.15 | 0.63 | 0.74 | 24 | 0 | 0 |
| | 0.14 | 0.79 | 0.63 | 99 | 0 | 0 |
| | 0.51 | 0.50 | 0.70 | 0 | 0 | 0 |
| | 0.22 | 0.24 | 0.57 | 26 | 25 | 0 |
| | 0.08 | 0.78 | 0.51 | 99 | 0 | 0 |
| | 0.46 | 0.56 | 0.52 | 0 | 0 | 0 |
| | 0.40 | 0.66 | 0.55 | 25 | 0 | 0 |
| | 0.52 | 0.37 | 0.58 | 0 | 25 | 0 |
| | 0.15 | 0.16 | 0.50 | 56 | 25 | 0 |
| Task 2 (U) | | | | | | |
| | 0.47 | 0.34 | 0.69 | 0 | 25 | 0 |
| | 0.08 | 0.64 | 0.86 | 99 | 0 | 0 |
| | 0.15 | 0.83 | 0.67 | 57 | 0 | 0 |
| | 0.53 | 0.59 | 0.74 | 0 | 0 | 0 |
| | 0.15 | 0.25 | 0.58 | 26 | 25 | 0 |
| | 0.08 | 0.67 | 0.77 | 98 | 0 | 0 |
| | 0.44 | 0.56 | 0.54 | 25 | 0 | 0 |
| | 0.43 | 0.67 | 0.74 | 25 | 0 | 0 |
| | 0.63 | 0.37 | 0.59 | 0 | 25 | 0 |
| | 0.14 | 0.16 | 0.48 | 53 | 25 | 0 |
| Task 3 | | | | | | |
| | 0.35 | 0.13 | 0.14 | 11 | 31 | 16 |
| | 0.26 | 0.15 | 0.15 | 33 | 49 | 40 |
| | 0.14 | 0.13 | 0.12 | 18 | 37 | 15 |
| | 0.11 | 0.25 | 0.33 | 20 | 16 | 16 |
| | 0.15 | 0.10 | 0.23 | 27 | 33 | 11 |
| | 0.14 | 0.06 | 0.13 | 13 | 25 | 18 |
| | 0.31 | 0.14 | 0.23 | 14 | 31 | 14 |
| | 0.15 | 0.17 | 0.13 | 27 | 17 | 37 |
| | 0.62 | 0.37 | 0.31 | 0 | 13 | 13 |
| | 0.28 | 0.18 | 0.26 | 12 | 46 | 12 |

**Table 4 Dissimilarity indices and length of spatially and temporally matched segments in Group 2.** U, upper arms; F, forearms; H, hands; columns 4–6 show the percentage of similarity shown between right and left arms based on minimum and maximum time indices of congruent segments

| | U EP-DI | F EP-DI | H EP-DI | U % of sim | F % of sim | H % of sim |
|---|---|---|---|---|---|---|
| Task 1 | | | | | | |
| | 0.03 | 0.44 | 0.79 | 97 | 33 | 0 |
| | 0.05 | 0.22 | 0.39 | 88 | 31 | 38 |
| | 0.03 | 0.04 | 0.09 | 96 | 96 | 96 |
| | 0.77 | 0.06 | 0.14 | 0 | 87 | 93 |
| | 0.01 | 0.07 | 0.05 | 33 | 78 | 89 |
| | 0.53 | 0.38 | 0.87 | 0 | 31 | 0 |
| | 0.33 | 0.36 | 0.18 | 29 | 21 | 29 |
| | 0.13 | 0.44 | 0.33 | 82 | 13 | 29 |
| | 0.05 | 0.04 | 0.31 | 86 | 86 | 29 |
| | 0.14 | 0.29 | 0.26 | 69 | 31 | 31 |
| Task 2 (B) | | | | | | |
| | 0.78 | 0.60 | 0.30 | 0 | 0 | 25 |
| | 0.28 | 0.61 | 0.55 | 25 | 0 | 0 |
| | 0.06 | 0.38 | 0.27 | 99 | 25 | 26 |
| | 0.23 | 0.18 | 0.84 | 25 | 25 | 0 |
| | 0.17 | 0.74 | 0.80 | 26 | 0 | 0 |
| | 0.35 | 0.33 | 0.77 | 24 | 26 | 0 |
| | 0.25 | 0.67 | 0.70 | 24 | 0 | 0 |
| | 0.15 | 0.68 | 0.87 | 29 | 0 | 0 |
| | 0.63 | 0.69 | 0.71 | 0 | 0 | 0 |
| | 0.16 | 0.32 | 0.43 | 25 | 25 | 25 |
| Task 2 (U) | | | | | | |
| | 0.81 | 0.61 | 0.33 | 0 | 0 | 25 |
| | 0.29 | 0.64 | 0.58 | 25 | 0 | 0 |
| | 0.06 | 0.43 | 0.37 | 100 | 25 | 25 |
| | 0.27 | 0.19 | 0.77 | 25 | 25 | 0 |
| | 0.19 | 0.87 | 0.87 | 26 | 0 | 0 |
| | 0.39 | 0.47 | 0.78 | 25 | 0 | 0 |
| | 0.40 | 0.59 | 0.84 | 25 | 0 | 0 |
| | 0.14 | 0.64 | 0.86 | 30 | 0 | 0 |
| | 0.64 | 0.70 | 0.68 | 0 | 0 | 0 |
| | 0.15 | 0.31 | 0.41 | 78 | 25 | 25 |
| Task 3 | | | | | | |
| | 0.34 | 0.09 | 0.13 | 25 | 30 | 40 |
| | 0.28 | 0.46 | 0.24 | 15 | 0 | 15 |
| | 0.23 | 0.22 | 0.18 | 13 | 13 | 13 |
| | 0.59 | 0.46 | 0.16 | 0 | 0 | 33 |
| | 0.53 | 0.08 | 0.14 | 0 | 31 | 14 |
| | 0.42 | 0.56 | 0.56 | 30 | 0 | 0 |
| | 0.32 | 0.24 | 0.23 | 20 | 20 | 20 |
| | 0.14 | 0.07 | 0.61 | 8 | 15 | 0 |
| | 0.14 | 0.60 | 0.25 | 30 | 0 | 14 |
| | 0.14 | 0.35 | 0.40 | 22 | 13 | 13 |

A

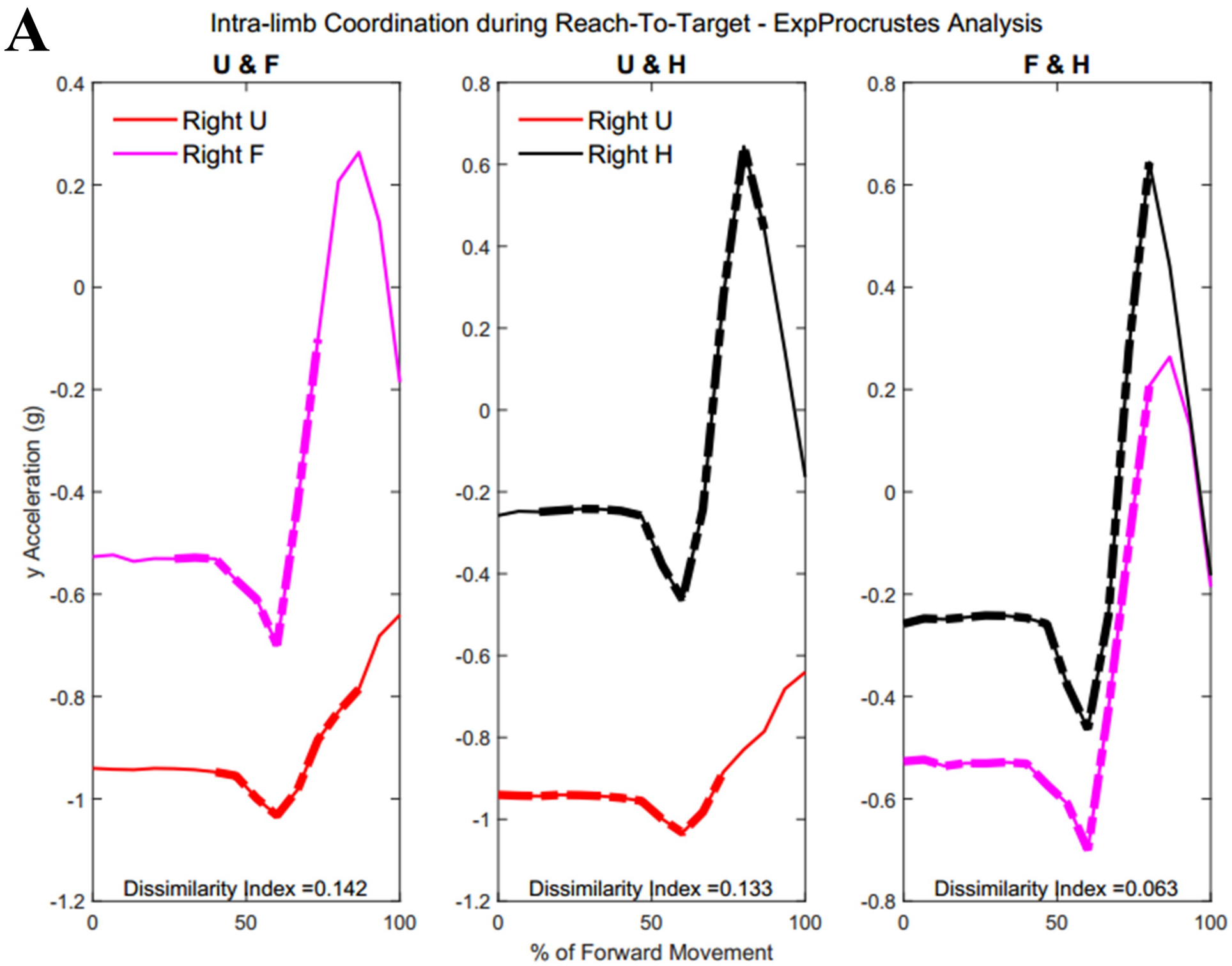

B

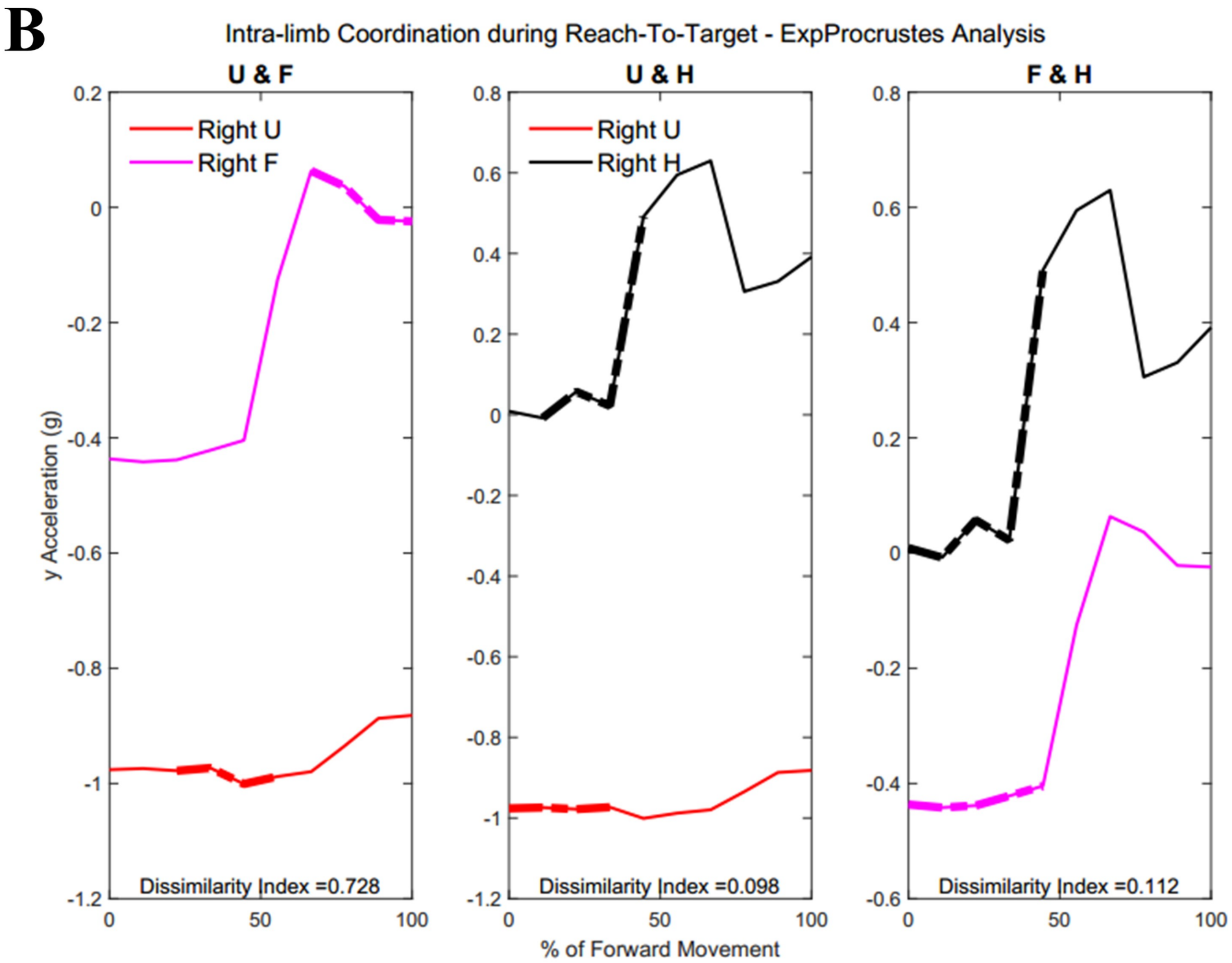

**Figure 13** (A) Group 1 and B) Group 2 figures of intra-limb coordination in the dominant limb during Task 1.

**Table 5 Intra-limb coordination: Dissimilarity indices between three joint pairs of the dominant limb in Group 1.** Group 1 had increased intra-limb coordination.

| | U & F EP-DI | U & H EP-DI | F & H EP-DI | U & F % of sim | U & H % of sim | F & H % of sim |
|---|---|---|---|---|---|---|
| Task 1 | | | | | | |
| | 0.12 | 0.29 | 0.13 | 0.0 | 0.0 | 80.0 |
| | 0.13 | 0.12 | 0.10 | 26.7 | 73.3 | 86.7 |
| | 0.27 | 0.20 | 0.11 | 0.0 | 0.0 | 45.5 |
| | 0.05 | 0.15 | 0.12 | 18.2 | 36.4 | 27.3 |
| | 0.14 | 0.13 | 0.06 | 46.7 | 73.3 | 80.0 |
| | 0.00 | 0.09 | 0.16 | 12.5 | 37.5 | 75.0 |
| | 0.10 | 0.10 | 0.54 | 0.0 | 0.0 | 0.0 |
| | 0.45 | 0.70 | 0.46 | 0.0 | 33.3 | 33.3 |
| | 0.09 | 0.16 | 0.01 | 0.0 | 25.0 | 25.0 |
| | 0.44 | 0.09 | 0.07 | 15.4 | 53.8 | 69.2 |
| Task 2 (B) | | | | | | |
| | 0.46 | 0.58 | 0.48 | 0.0 | 0.0 | 25.0 |
| | 0.69 | 0.68 | 0.79 | 24.5 | 0.0 | 24.5 |
| | 0.52 | 0.82 | 0.91 | 24.8 | 0.0 | 0.0 |
| | 0.44 | 0.39 | 0.62 | 24.5 | 24.5 | 0.0 |
| | 0.63 | 0.60 | 0.43 | 25.2 | 0.0 | 25.2 |
| | 0.53 | 0.77 | 0.80 | 25.0 | 25.0 | 25.0 |
| | 0.58 | 0.53 | 0.59 | 25.3 | 0.0 | 25.3 |
| | 0.85 | 0.59 | 0.66 | 0.0 | 26.6 | 25.0 |
| | 0.64 | 0.49 | 0.37 | 25.0 | 0.0 | 25.0 |
| | 0.28 | 0.43 | 0.41 | 24.6 | 0.0 | 24.6 |
| Task 2 (U) | | | | | | |
| | 0.46 | 0.56 | 0.49 | 0.0 | 0.0 | 24.8 |
| | 0.70 | 0.77 | 0.80 | 24.7 | 0.0 | 24.7 |
| | 0.53 | 0.84 | 0.91 | 24.7 | 0.0 | 0.0 |
| | 0.48 | 0.41 | 0.58 | 24.8 | 24.8 | 24.8 |
| | 0.59 | 0.58 | 0.46 | 25.0 | 0.0 | 25.0 |
| | 0.56 | 0.77 | 0.80 | 25.0 | 25.0 | 25.0 |
| | 0.60 | 0.63 | 0.58 | 24.9 | 0.0 | 24.9 |
| | 0.60 | 0.75 | 0.67 | 25.2 | 25.2 | 25.2 |
| | 0.68 | 0.46 | 0.38 | 25.0 | 25.0 | 25.0 |
| | 0.29 | 0.36 | 0.42 | 24.5 | 0.0 | 24.5 |

same relative velocities and stability (Group 1: 22.38 ± 7.87%, Group 2: 12.48 ± 13.16%, $p = 0.028$).

Figure 15 demonstrates intra-limb coordination in the non-dominant limb during the cabinet task. Groups differed most in the coordination between the upper arm and forearm segments of the non-dominant arm, with Group 1 showing greater percentages of

**Table 6** **Intra-limb coordination: dissimilarity indices between three joint pairs of the dominant limb in Group 2.**

| | U & F EP-DI | U & H EP-DI | F & H EP-DI | U & F % of sim | U & H % of sim | F & H % of sim |
|---|---|---|---|---|---|---|
| Task 1 | | | | | | |
| | 0.20 | 0.12 | 0.80 | 0.0 | 90.9 | 0.0 |
| | 0.40 | 0.46 | 0.17 | 18.8 | 0.0 | 31.3 |
| | 0.20 | 0.48 | 0.27 | 26.1 | 30.4 | 30.4 |
| | 0.59 | 0.69 | 0.32 | 20.0 | 0.0 | 33.3 |
| | 0.73 | 0.10 | 0.11 | 0.0 | 33.3 | 44.4 |
| | 0.29 | 0.49 | 0.76 | 0.0 | 31.3 | 0.0 |
| | 0.55 | 0.17 | 0.85 | 0.0 | 29.4 | 0.0 |
| | 0.55 | 0.17 | 0.85 | 0.0 | 29.4 | 0.0 |
| | 0.14 | 0.45 | 0.11 | 0.0 | 0.0 | 64.3 |
| | 0.01 | 0.16 | 0.29 | 15.4 | 0.0 | 30.8 |
| Task 2 (B) | | | | | | |
| | 0.77 | 0.60 | 0.79 | 25.2 | 25.2 | 25.2 |
| | 0.51 | 0.57 | 0.61 | 25.1 | 0.0 | 0.0 |
| | 0.46 | 0.41 | 0.48 | 25.0 | 25.0 | 25.0 |
| | 0.63 | 0.76 | 0.64 | 25.0 | 0.0 | 25.0 |
| | 0.91 | 0.66 | 0.64 | 0.0 | 25.3 | 25.3 |
| | 0.65 | 0.78 | 0.54 | 0.0 | 0.0 | 24.3 |
| | 0.69 | 0.82 | 0.75 | 0.0 | 0.0 | 24.2 |
| | 0.36 | 0.30 | 0.79 | 0.0 | 0.0 | 0.0 |
| | 0.71 | 0.72 | 0.70 | 0.0 | 0.0 | 25.8 |
| | 0.81 | 0.66 | 0.74 | 24.9 | 24.9 | 24.9 |
| Task 2 (U) | | | | | | |
| | 0.76 | 0.64 | 0.79 | 25.1 | 25.1 | 25.1 |
| | 0.53 | 0.57 | 0.63 | 25.1 | 0.0 | 0.0 |
| | 0.53 | 0.41 | 0.55 | 25.1 | 25.1 | 25.1 |
| | 0.65 | 0.78 | 0.70 | 24.7 | 24.7 | 24.7 |
| | 0.87 | 0.66 | 0.61 | 0.0 | 25.0 | 25.0 |
| | 0.69 | 0.77 | 0.65 | 0.0 | 25.1 | 25.1 |
| | 0.53 | 0.90 | 0.82 | 0.0 | 0.0 | 24.8 |
| | 0.40 | 0.41 | 0.79 | 0.0 | 0.0 | 0.0 |
| | 0.66 | 0.75 | 0.76 | 0.0 | 0.0 | 24.7 |
| | 0.76 | 0.64 | 0.71 | 24.8 | 24.8 | 24.8 |

similarity in the non-dominant limb as well (left upper arm and forearm, Group 1: 28.67 ± 14.59%, Group 2: 30.84 ± 18.21%, $p = 0.083$).

Tables 7 and 8 show dissimilarity indices and percentages of similarity for all subjects in Group 1 and Group 2, respectively. In summary, both groups showed greater coordination during the first task, particularly between the forearm and hand segments. Groups did not differ significantly during the buttoning and unbuttoning tasks. Group 1 demonstrated a

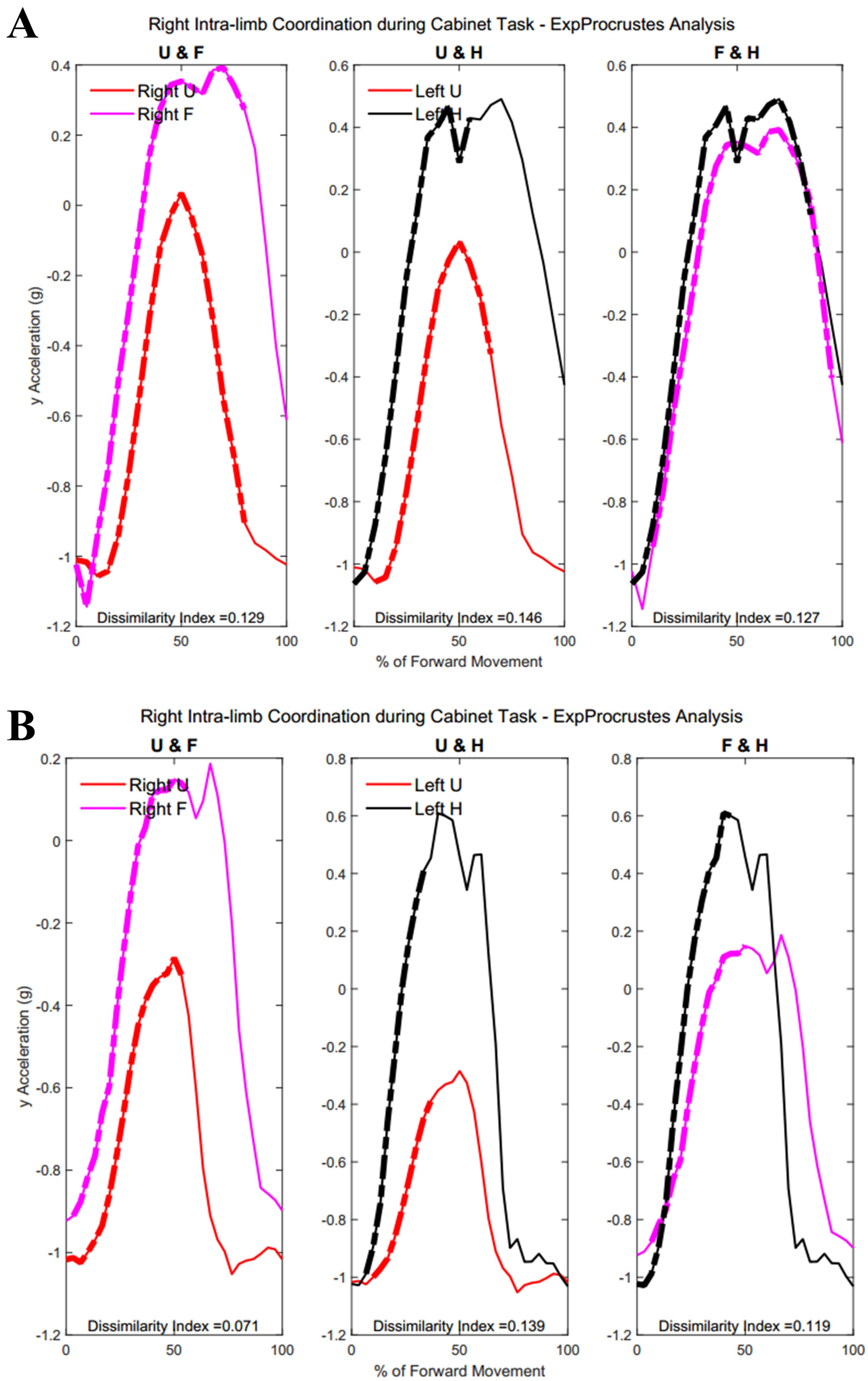

**Figure 14** (A) Group 1 and (B) Group 2 figures of intra-limb coordination in the dominant limb during Task 3.

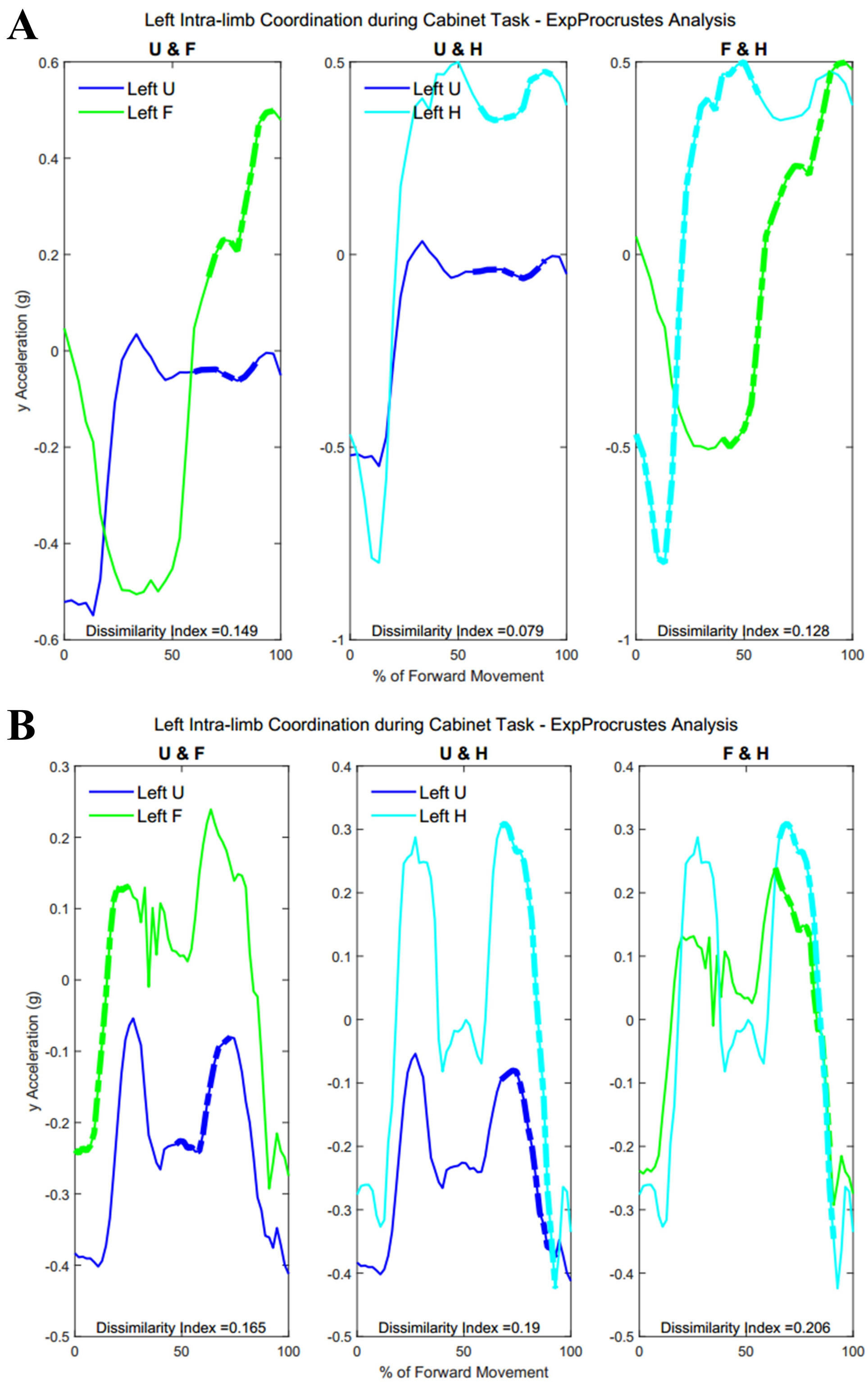

**Figure 15** (A) Group 1 and (B) Group 2 figures of intra-limb coordination in the non-dominant limb during Task 3.

**Table 7 Dominant and non-dominant limb dissimilarities and percentages of similarity in Group 1.** Group 1 demonstrated significant and greater intra-limb coordination between the forearm and hand segments in both limbs.

| | U & F EP-DI | U & H EP-DI | F & H EP-DI | U & F % of sim | U & H % of sim | F & H % of sim |
|---|---|---|---|---|---|---|
| Task 3 (R) | | | | | | |
| | 0.20 | 0.27 | 0.13 | 0.0 | 25.0 | 45.0 |
| | 0.12 | 0.20 | 0.05 | 48.3 | 25.0 | 60.0 |
| | 0.11 | 0.21 | 0.13 | 43.3 | 23.3 | 76.7 |
| | 0.10 | 0.12 | 0.13 | 0.0 | 46.7 | 43.3 |
| | 0.33 | 0.26 | 0.09 | 20.0 | 25.0 | 0.0 |
| | 0.15 | 0.51 | 0.12 | 0.0 | 0.0 | 40.0 |
| | 0.13 | 0.14 | 0.56 | 0.0 | 30.0 | 0.0 |
| | 0.13 | 0.15 | 0.13 | 80.0 | 55.0 | 85.0 |
| | 0.17 | 0.21 | 0.19 | 20.0 | 30.0 | 25.0 |
| | 0.15 | 0.14 | 0.08 | 53.3 | 26.7 | 53.3 |
| Task 3 (L) | | | | | | |
| | 0.18 | 0.23 | 0.14 | 24.4 | 0.0 | 26.7 |
| | 0.15 | 0.14 | 0.32 | 35.6 | 26.7 | 24.4 |
| | 0.28 | 0.18 | 0.13 | 0.0 | 25.5 | 0.0 |
| | 0.20 | 0.12 | 0.51 | 0.0 | 64.4 | 24.4 |
| | 0.31 | 0.15 | 0.14 | 24.4 | 57.8 | 60.0 |
| | 0.15 | 0.24 | 0.17 | 55.0 | 25.0 | 25.0 |
| | 0.24 | 0.31 | 0.42 | 0.0 | 25.7 | 25.7 |
| | 0.15 | 0.08 | 0.13 | 30.0 | 33.3 | 0.0 |
| | 0.39 | 0.37 | 0.15 | 0.0 | 25.0 | 60.0 |
| | 0.07 | 0.16 | 0.15 | 0.0 | 25.0 | 0.0 |

significant difference in upper arm and forearm coordination in both the dominant and the non-dominant limb during the cabinet task.

## Statistical findings

Table 9 details the two-way Analysis of Variance (ANOVA) statistical evaluation of the influence of task demands and age on the variance in inter-limb coordination scores. Coordination scores were calculated for each subject's inter- and intra-limb coordination by dividing the percentage of similarity by the dissimilarity index of spatio-temporally coordinating segments. Therefore high percentages of similarity and low dissimilarity scores both contributed to a higher coordination score. In the case of inter-limb coordination of the upper arms and the forearms, age contributed to the variance in scores across groups. Age and task had a significant combined effect on the inter-limb coordination demonstrated in all tasks by the hand segments.

Table 10 details the two-way ANOVA statistical evaluation of the influence of task and age on intra-limb coordination scores. Age only had a significant influence on the variance of forearm scores, while task had a significant influence on the variance of hand scores. Assessment of the Quantile-Quantile plots of the residuals generated by

**Table 8 Dominant and non-dominant limb dissimilarities and percentages of similarity in Group 2.**

| | U & F EP-DI | U & H EP-DI | F & H EP-DI | U & F % of sim | U & H % of sim | F & H % of sim |
|---|---|---|---|---|---|---|
| Task 3 (R) | | | | | | |
| | 0.09 | 0.35 | 0.40 | 25.0 | 25.0 | 0.0 |
| | 0.13 | 0.31 | 0.15 | 0.0 | 0.0 | 37.5 |
| | 0.36 | 0.14 | 0.26 | 0.0 | 53.3 | 23.3 |
| | 0.13 | 0.36 | 0.12 | 37.5 | 27.5 | 95.0 |
| | 0.54 | 0.46 | 0.26 | 0.0 | 0.0 | 25.0 |
| | 0.19 | 0.14 | 0.15 | 20.0 | 26.0 | 62.0 |
| | 0.15 | 0.25 | 0.26 | 60.0 | 23.3 | 26.7 |
| | 0.15 | 0.89 | 0.24 | 0.0 | 0.0 | 0.0 |
| | 0.11 | 0.52 | 0.38 | 33.3 | 0.0 | 0.0 |
| | 0.07 | 0.14 | 0.12 | 53.3 | 26.7 | 43.3 |
| Task 3 (L) | | | | | | |
| | 0.29 | 0.11 | 0.21 | 25.0 | 60.0 | 25.0 |
| | 0.61 | 0.26 | 0.48 | 0.0 | 24.6 | 13.8 |
| | 0.16 | 0.63 | 0.47 | 25.5 | 0.0 | 0.0 |
| | 0.65 | 0.15 | 0.33 | 23.3 | 60.0 | 0.0 |
| | 0.45 | 0.22 | 0.12 | 25.7 | 25.7 | 100.0 |
| | 0.08 | 0.14 | 0.11 | 47.5 | 35.0 | 55.0 |
| | 0.49 | 0.52 | 0.33 | 0.0 | 0.0 | 0.0 |
| | 0.23 | 0.42 | 0.29 | 21.7 | 24.4 | 26.8 |
| | 0.28 | 0.17 | 0.13 | 0.0 | 24.0 | 0.0 |
| | 0.17 | 0.19 | 0.21 | 0.0 | 25.5 | 25.5 |

two-way ANOVA demonstrated a bimodal deviation of coordination scores from a normal distribution, with slightly more extreme values than would be expected. Figure 16 shows the Quantile-Quantile plots for interlimb coordination score residuals for the three limb segments.

## DISCUSSION

Traditional methods of showing bilateral synergistic movement or coordination between the two limbs show kinematic quantities that change in a correlated manner, *e.g.*, by assessing the force generation of both hands during a given task. Not only is such methodology limiting in the movements that can be assessed, but a correlated change in a particular number of variables does not necessarily indicate a time-dependent coordination of movement. In this article, we proposed the use of accelerometry to identify submovements or sub-trajectories that spatially and temporally proceed in the same shape, implying a similar progression rather than discrete changes that may occur out of phase with one another. In order to evaluate both the use of accelerometers and the novel procrustean segmentation methodology, this study was limited in only assessing basic reaching tasks. However, we expect it can be adapted to any movement requiring both

**Table 9 Two-way ANOVA of inter-limb coordination scores.** Asterisks highlight values that were found to be below the chosen statistical significance level of 0.05.

| | | | | | (Upper Arm) |
|---|---|---|---|---|---|
| Source | Sum Sq. | d.f. | Mean Sq. | F | Prob >F |
| Age | 10.3446 | 3 | 3.44821 | 2.91 | 0.0401* |
| Task | 0.0008 | 1 | 0.00079 | 0 | 0.9794 |
| Age:Task | 0.8755 | 3 | 0.29182 | 0.25 | 0.8635 |
| Error | 85.2032 | 72 | 1.18338 | | |
| Total | 96.4241 | 79 | | | |
| | | | | | (Forearm) |
| Source | Sum Sq. | d.f. | Mean Sq. | F | Prob >F |
| Age | 43.114 | 3 | 14.3714 | 18.41 | 0* |
| Task | 2.185 | 1 | 2.1853 | 2.8 | 0.0986 |
| Age:Task | 2.97 | 3 | 0.9901 | 1.27 | 0.2917 |
| Error | 56.202 | 72 | 0.7806 | | |
| Total | 104.472 | 79 | | | |
| | | | | | (Hand) |
| Source | Sum Sq. | d.f. | Mean Sq. | F | Prob >F |
| Age | 64.128 | 3 | 21.3759 | 35.21 | 0* |
| Task | 0 | 1 | 0 | 0 | 0.9963 |
| Age:Task | 6.567 | 3 | 2.1889 | 3.61 | 0.0174* |
| Error | 43.717 | 72 | 0.6072 | | |
| Total | 114.411 | 79 | | | |

upper extremities. Since the method uses the entire movement set and segments it based on task features, it can be applied to any set of clinically produced data to find trajectories that are highly similar in part to control trajectories.

The expected age-related differences in bilateral coordination include a decreased average velocity and reduced smoothness of the whole arm, to reflect slower and less accurate ability to replicate coordinated movements (*Maes et al., 2017*; *Roman-Liu & Tokarski, 2020*; *Roman-Liu & Mockałło, 2020*). In addition, prior literature demonstrates that as individuals age, the upper limbs increase in symmetrical performance, perhaps due to loss of motor lateralization or through motor adaptation to age-related muscle weakness (*Pan et al., 2023*; *King et al., 2017*). In Fig. 8, the accelerometry data was found sufficient to demonstrate significant differences between groups in the peak angular velocities of the right forearm and left upper arm throughout all tasks. All three tasks require similar initial forward progression of the upper arm and forearm segments. In Fig. 9, the rotation of the arm about the forward axis was shown as a representation of movement smoothness, with significant differences between groups specifically in the right and left upper arm segments. Interestingly, while age-related differences were seen between groups in both peak velocity and peak roll angle, since they were calculations of individual limb segments, they do not correspond entirely with whole arm observations. The group with a higher mean age had

**Table 10** **Two-way ANOVA of intra-limb coordination.** Asterisks highlight values that were found to be below the chosen statistical significance level of 0.05.

| | | | | | (U.Arm - F.Arm) |
|---|---|---|---|---|---|
| Source | Sum Sq. | d.f. | Mean Sq. | F | Prob >F |
| Age | 0.909 | 3 | 0.30316 | 0.21 | 0.8905 |
| Task | 5.18 | 1 | 5.17967 | 3.56 | 0.0625 |
| Age:Task | 3.297 | 3 | 1.09883 | 0.75 | 0.5225 |
| Error | 133.976 | 92 | 1.45626 | | |
| Total | 141.541 | 99 | | | |
| | | | | | (U.Arm - H) |
| Source | Sum Sq. | d.f. | Mean Sq. | F | Prob >F |
| Age | 25.422 | 3 | 8.47411 | 9.31 | 0* |
| Task | 0.215 | 1 | 0.21535 | 0.24 | 0.6278 |
| Age:Task | 2.856 | 3 | 0.95202 | 1.05 | 0.3761 |
| Error | 83.733 | 92 | 0.91014 | | |
| Total | 112.771 | 99 | | | |
| | | | | | (F.Arm - H) |
| Source | Sum Sq. | d.f. | Mean Sq. | F | Prob >F |
| Age | 5.0251 | 3 | 1.67503 | 1.77 | 0.1592 |
| Task | 4.3808 | 1 | 4.38083 | 4.62 | 0.0343* |
| Age:Task | 3.3511 | 3 | 1.11705 | 1.18 | 0.3226 |
| Error | 87.2631 | 92 | 0.94851 | | |
| Total | 99.9658 | 99 | | | |

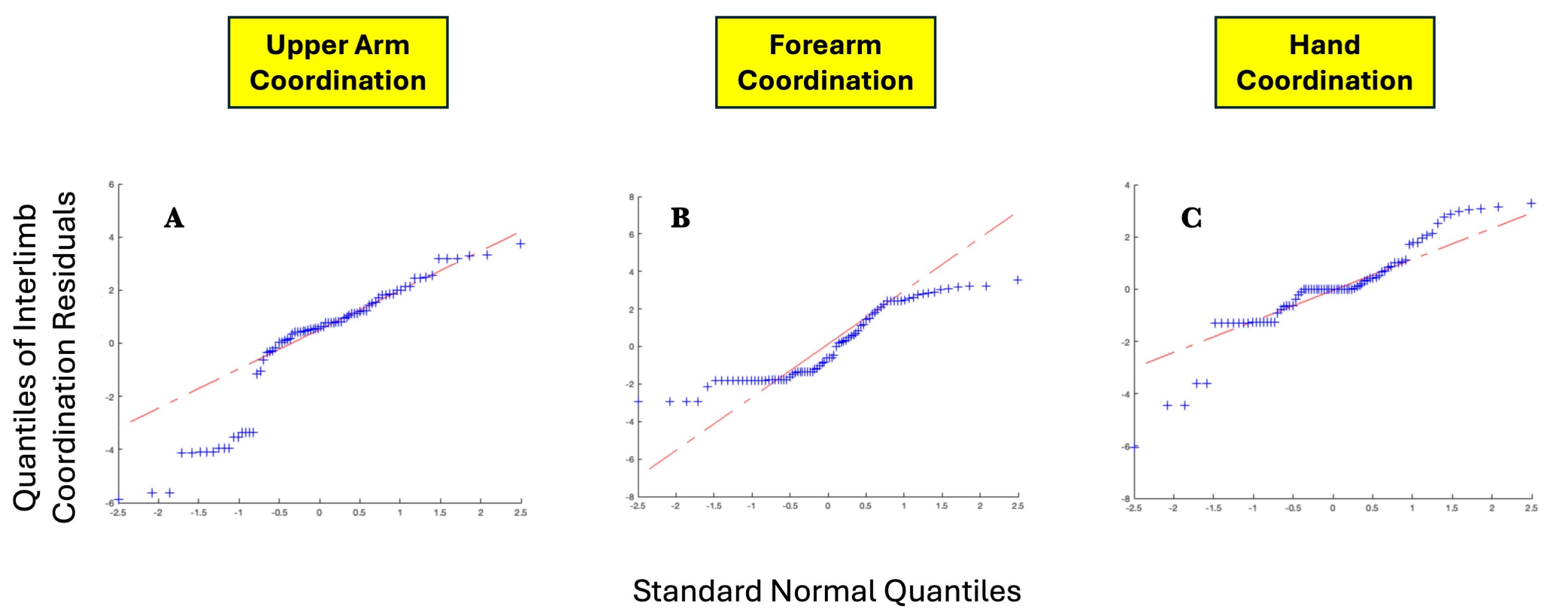

**Figure 16** **QQ plots for (A) upper arm interlimb coordination residuals, (B) forearm interlimb coordination, and C hand interlimb coordination.**

a greater peak velocity of the upper arms, indicating a limitation in using peak or average velocity as a sole metric of movement speed.

The remaining hypotheses concerned bilateral coordination differences between groups while performing tasks of increasing complexity and decreasing symmetry. We expected Group 1 to demonstrate greater intra-limb coordination, and Group 2 to demonstrate

longer periods of inter-limb coordination. The first of these hypotheses were generated based on prior literature demonstrated more well-defined bursts of motor activation in younger individuals (*Monaco, Ghionzoli & Micera, 2010*). While aging does not significantly alter the presence or number of muscle synergies, aging and disability can lead to alterations of the neural control strategy, implicating there may be compensatory adaptations with age reflected in coordination, such as increased mechanical expenditure of limb segments and reduced biomechanical capacities (*Guo et al., 2022*; *Dussault-Picard et al., 2024*). Due to the non-periodic and motor abundant nature of the upper extremity, we expected to see such adaptation of motor control to be reflected in the length and presence of congruent trajectories between the two arms. These hypotheses also took into account the initiation and completion of the specific bimanual tasks investigated in this study. The achievement of coordinated symmetry between the upper extremities is task specific, the result of the motor demands imposed during each specific motor task (*Teixeira, 2008*), and therefore task complexity was expected to impact coordination scores between groups (*Bujas et al., 2018*). Motor practice and muscle memory collected over the lifetime was also expected to impact motor performance of the group with higher mean age. Finally, these hypotheses also incorporated findings of reduced asymmetry and increased mirroring between limbs after the age of 30 years, expecting longer periods of inter-limb coordination in Group 2 (*Koerte et al., 2010*; *Knights, Morcom & Henson, 2021*; *Kalisch et al., 2006*).

The findings that differ between groups are intriguing and trend with intuitive understandings of how the upper limbs are coordinated in daily tasks. The group consisting of subjects above the age of thirty-five were indeed found to demonstrate more inter-limb coordination, which may be reduction in motor laterality with age, or due to greater motor performance memory of the specific tasks. Interestingly, Group 2 were more familiar with the buttoning task, while Group 1 reported no familiarity with buttoning a shirt placed in front, such as one may do while assisting another individual with dressing. The group below the age of thirty was found to demonstrate longer periods of intra-limb coordination, or coordination between segments of the same arm, even during a task with independent asymmetry in movement. Group 1 demonstrated greatest intra-limb coordination between each of the three limb segment pairs, hand-forearm, hand-upper arm, and forearm-upper arm, throughout Task 1, and during the initiation of Tasks 2 and 3. Group 2 showed greater variability in how individuals chose to perform the more asymmetric tasks. Further, congruence in intra-limb coordination was found also in the non-dominant limb during Task 3 for both groups.

The previously developed Reach Severity and Dissimilarity Index (RSDI) must be altered further to be sensitive to age-differences in contrast to impairment severity in the presence of functional limitations. The findings of greater inter-limb coordination in Group 2 and greater intra-limb coordination in Group 1 introduces many more interesting questions for future study. Of interest are the underlying mechanisms driving coordination to occur at particular phases of movement in either group. In addition to task complexity and age, movement patterns may be impacted by muscle learning, lifetime motor practice, and the relative novelty of tasks. Another limitation is the need to validate the findings of this study by matching it to muscle activation patterns to explore the specific progression

of muscle recruitment during the coordination shown. In future studies, it would be invaluable to study coordination on a neural and mechanistic level, in addition to the behavioral analysis presented here. We hope the methodology described in this article is a step towards human activity recognition technologies to be harnessed to create low-cost, individualized therapies and monitoring systems that reduce patient burden, standardize clinical practices and objective scores, and ultimately reduce the impact of functional limitations in movement on the incidence of disability.

## CONCLUSIONS

Human activity recognition (HAR) refers to a vast variety of computational methodologies concerned with collecting movement data, processing and segmenting it, and identifying characteristic movements in order to qualify motor performance. Recent advancements in wearable sensor technologies have made it more possible for healthcare monitoring and rehabilitative interventions to be carried out remotely, alleviating patient burden and reducing the potential for non-compliance (*Ranasinghe, Al Machot & Mayr, 2016*; *Chae et al., 2020*; *Avci et al., 2010*).

Acceleration sensors or inertial measurement units (IMUs) that combine accelerometers with gyroscopes have gained popularity as wearable sensors due to a high level of resolution and accuracy compared to other approaches (*Janidarmian et al., 2017*). Although many sensors suffer from high levels of noise, differences in precision, and various biases, the Expanding Procrustes methodology is able to segment and characterize movement features with relative simplicity from accelerometer and gyroscope data. Sensor-produced movement data is frequently segmented using fixed-sliding-window in order to find characteristics in small segments of data that can then be evaluated in order to extract further information and repeated movements. This article borrows the fixed-sliding-window method of segmentation and combines it with Procrustean distance analysis in order to identify segments that are spatially and temporally congruent between limbs and limb segments. Between high resolution wearable sensors and a statistical shape analysis applied across curve similarities to select dissimilarities, the expanding Procrustes method removes the need to filter the data. Previous methodologies have also validated the ability for task-specific features of daily movements to be identified from wearable inertial sensors (*Hsu et al., 2018*; *Hussain, Alt Murphy & Sunnerhagen, 2018*).

The upper extremity, particularly our hands, play a crucial role in daily activities and quality of life. With age, the upper extremity undergoes a number of physiological changes that must be evident in any effective movement data collection paradigm. Changes may take place in upper extremity motor control due to muscle changes or age related differences in hand laterality or dominance (*Papadatou-Pastou et al., 2020*; *Khanafer et al., 2021*). Any study of the motor abundant and non-periodic upper extremity is incomplete without assessing the impact of bilateral movement coordination. While there are many previous studies addressing age-related differences in hand function, there is yet limited understanding of how coordination changes with age and task complexity (*Sebastjan et al., 2017*).

The proposed study offers significant potential for understanding bilateral synergistic movement between the joints and muscles of the upper extremity. There is currently no consensus amongst clinicians and movement researchers in which metrics compose a standard and comprehensive index of upper extremity motor performance. Objective measures of motor performance are of particular interest to create individual rehabilitation strategies for individuals with varying degrees of functional limitations and movement impairments of the upper extremity.

Bilateral synergistic movement is an invaluable feature of the upper extremities, making a wide variety of movements possible. This study aimed to validate the use of accelerometers to extract task-specific features, extract age-related differences, and calculate coordination quality scores. Dysregulation of bilateral coordination is also a sequela of neural injuries leading to functional limitations in movement, such as those arising after stroke (*Cirstea & Levin, 2000*; *Oosterwijk et al., 2018*). In order to comprehensively evaluate the methods developed in this article, the wearable system, the data segmentation strategies, and the Procrustean analysis must be applied in the future to clinical conditions such as Parkinson's, stroke, multiple sclerosis, and traumatic brain injuries. We hope such future applications can contribute to a greater understanding of the upper extremity and how its motor abundance can be harnessed to inform rehabilitative interventions.

## ACKNOWLEDGEMENTS

We would like to extend our appreciation to all the participants who generously volunteered their time and effort to this study. Further, we are grateful to Dr. Kathryn Laskey for guiding the analysis and writing of this study.

## APPENDIX

**Table A1 Maximum angular velocities in the positive y-direction for Group 1 and Group 2.** Asterisks highlight values that were found to be below the chosen statistical significance level of 0.05.

| | RU max (°/s) | RF max* (°/s) | RH max (°/s) | LU max (°/s) | LF max (°/s) | LH max (°/s) |
|---|---|---|---|---|---|---|
| G1.01 | 50.0 | 113.2 | 128.8 | 70.7 | 167.8 | 156.5 |
| G1.02 | 70.7 | 122.5 | 178.9 | 76.4 | 127.0 | 305.8 |
| G1.03 | 90.0 | 150.2 | 219.8 | 81.4 | 130.6 | 258.4 |
| G1.04 | 94.7 | 128.4 | 366.9 | 101.4 | 340.2 | 166.1 |
| G1.05 | 152.8 | 191.9 | 204.0 | 159.5 | 232.4 | 210.7 |
| G1.06 | 134.4 | 141.6 | 207.9 | 105.9 | 125.6 | 264.3 |
| G1.07 | 98.9 | 216.4 | 219.0 | 112.2 | 170.4 | 253.5 |
| G1.08 | 145.7 | 139.0 | 225.4 | 101.1 | 166.0 | 246.8 |
| G1.09 | 232.2 | 319.8 | 924.7 | 353.8 | 306.9 | 689.6 |
| G1.10 | 128.1 | 150.6 | 405.1 | 110.2 | 106.9 | 354.8 |
| Mean | 119.8 | 167.4 | 308.1 | 127.2 | 187.4 | 290.6 |
| STD | 51.5 | 62.1 | 232.2 | 83.4 | 80.3 | 152.2 |

| | RU max (°/s) | RF max* (°/s) | RH max (°/s) | LU max (°/s) | LF max (°/s) | LH max (°/s) |
|---|---|---|---|---|---|---|
| G2.01 | 352.8 | 239.0 | 158.7 | 239.4 | 210.9 | 141.5 |
| G2.02 | 144.6 | 185.9 | 244.8 | 181.0 | 116.0 | 281.0 |
| G2.03 | 80.6 | 161.4 | 208.3 | 73.7 | 138.2 | 313.7 |
| G2.04 | 182.9 | 286.1 | 390.7 | 249.7 | 274.0 | 344.0 |
| G2.05 | 92.4 | 189.4 | 211.6 | 102.3 | 185.6 | 182.8 |
| G2.06 | 123.5 | 288.6 | 448.4 | 135.6 | 202.5 | 341.7 |
| G2.07 | 169.4 | 237.1 | 312.6 | 122.1 | 164.6 | 412.0 |
| G2.08 | 97.8 | 135.5 | 250.9 | 179.4 | 104.2 | 286.4 |
| G2.09 | 94.4 | 264.5 | 203.2 | 66.0 | 108.9 | 225.2 |
| G2.10 | 97.8 | 277.6 | 400.8 | 82.8 | 161.1 | 249.3 |
| Mean | 143.6 | 226.5 | 283.0 | 143.2 | 166.6 | 277.8 |
| STD | 81.4 | 55.0 | 99.3 | 66.6 | 53.5 | 81.0 |

**Table A2 Peak roll angle of limb segments for Groups 1 and 2.** Asterisks highlight values that were found to be below the chosen statistical significance level of 0.05.

| | RU° | RF° | RH° | LU°* | LF° | LH° |
|---|---|---|---|---|---|---|
| G1.01 | 19.3 | 53.0 | 88.6 | 8.9 | 0.4 | 2.8 |
| G1.02 | 30.2 | 54.7 | 89.6 | 9.5 | 0.1 | −8.7 |
| G1.03 | 59.8 | 89.2 | 82.5 | 11.7 | −131 | 4.6 |
| G1.04 | 54.3 | 51.2 | 86.7 | 8.5 | 5.2 | −1.0 |
| G1.05 | 34.5 | 58.5 | 82.6 | 11.6 | −5.9 | 11.4 |
| G1.06 | 47.6 | 42.5 | 89.0 | 6.6 | −6.4 | 7.8 |
| G1.07 | 22.5 | 89.3 | 88.8 | 10.1 | 12.3 | 18.5 |
| G1.08 | 34.0 | 59.4 | 88.7 | −5.2 | −149 | 17.7 |
| G1.09 | 32.1 | 63.8 | 87.3 | 9.9 | −0.8 | 16.5 |
| G1.10 | 28.3 | 62.0 | 89.1 | −5.8 | −3.7 | 14.2 |
| Mean | 36.3 | 62.4 | 87.3 | 6.6 | −2.7 | 8.4 |
| STD | 13.4 | 15.4 | 2.6 | 6.5 | 8.1 | 9.0 |
| G2.01 | 44.5 | 83.4 | 54.3 | 83.7 | 58.0 | 45.1 |
| G2.02 | 16.3 | 16.1 | 88.0 | 0.0 | 22.2 | 4.4 |
| G2.03 | 38.6 | 55.2 | 88.8 | 3.2 | −181 | 10.3 |
| G2.04 | 38.0 | 84.9 | 86.2 | 23.3 | −204 | 11.7 |
| G2.05 | 32.2 | 54.0 | 73.4 | 8.3 | −138 | 8.9 |
| G2.06 | 12.8 | 59.1 | 89.6 | 46.1 | 26.0 | 18.1 |
| G2.07 | 32.6 | 80.0 | 87.5 | 13.7 | −4.5 | 13.6 |
| G2.08 | 38.6 | 69.8 | 81.8 | 34.7 | −161 | 12.0 |
| G2.09 | 18.2 | 37.6 | 89.8 | 16.2 | 7.2 | 22.2 |
| G2.10 | 27.3 | 36.2 | 82.5 | 6.2 | −7.9 | 13.0 |
| Mean | 29.9 | 57.6 | 82.2 | 23.5 | 3.3 | 15.9 |
| STD | 10.9 | 22.8 | 11.0 | 25.6 | 25.3 | 11.3 |

**Table A3** Inter-limb coordination - traditional procrustes dissimilarity indices.

| | U | | | F | | | H | | |
|---|---|---|---|---|---|---|---|---|---|
| | T1 | T2 | T3 | T1 | T2 | T3 | T1 | T2 | T3 |
| G1.01 | 0.09 | 0.12 | 0.72 | 0.31 | 0.48 | 0.60 | 0.58 | 0.63 | 0.72 |
| G1.02 | 0.13 | 0.05 | 0.73 | 0.97 | 0.97 | 0.68 | 0.81 | 0.52 | 0.73 |
| G1.03 | 0.26 | 0.09 | 0.84 | 0.69 | 0.87 | 0.99 | 0.54 | 0.81 | 0.84 |
| G1.04 | 0.35 | 0.17 | 0.70 | 0.75 | 0.63 | 0.73 | 0.85 | 0.98 | 0.72 |
| G1.05 | 0.14 | 0.48 | 0.75 | 0.20 | 0.57 | 0.63 | 0.91 | 0.61 | 0.73 |
| G1.06 | 0.28 | 0.11 | 0.74 | 0.90 | 0.80 | 0.67 | 0.69 | 0.65 | 0.80 |
| G1.07 | 0.69 | 0.22 | 0.66 | 0.50 | 0.98 | 0.89 | 0.58 | 0.67 | 0.63 |
| G1.08 | 0.19 | 0.16 | 0.59 | 0.91 | 0.93 | 0.80 | 0.85 | 0.87 | 0.68 |
| G1.09 | 0.08 | 0.09 | 0.85 | 0.48 | 0.65 | 0.94 | 0.94 | 0.58 | 0.69 |
| G1.10 | 0.26 | 0.15 | 0.54 | 0.48 | 0.42 | 0.49 | 0.71 | 0.60 | 0.55 |
| Mean | 0.25 | 0.16 | 0.71 | 0.62 | 0.73 | 0.74 | 0.75 | 0.69 | 0.71 |
| STD | 0.18 | 0.12 | 0.10 | 0.27 | 0.21 | 0.16 | 0.15 | 0.14 | 0.080 |
| G2.01 | 0.81 | 0.60 | 0.73 | 1.00 | 0.92 | 0.62 | 0.84 | 0.08 | 0.63 |
| G2.02 | 0.16 | 0.39 | 0.53 | 0.55 | 0.99 | 0.97 | 0.94 | 0.71 | 0.58 |
| G2.03 | 0.04 | 0.06 | 0.78 | 0.10 | 0.36 | 0.73 | 0.35 | 0.46 | 0.69 |
| G2.04 | 0.22 | 0.55 | 0.76 | 0.71 | 0.82 | 0.79 | 0.68 | 0.58 | 0.87 |
| G2.05 | 0.18 | 0.15 | 0.57 | 0.42 | 0.87 | 0.50 | 0.72 | 0.91 | 0.59 |
| G2.06 | 0.19 | 0.16 | 0.59 | 0.91 | 0.93 | 0.80 | 0.85 | 0.87 | 0.68 |
| G2.07 | 0.14 | 0.52 | 0.56 | 0.97 | 0.96 | 0.53 | 0.49 | 0.64 | 0.70 |
| G2.08 | 0.14 | 0.18 | 0.90 | 0.97 | 0.93 | 1.00 | 0.49 | 0.95 | 0.90 |
| G2.09 | 0.31 | 0.18 | 0.84 | 0.93 | 0.65 | 0.99 | 0.79 | 0.70 | 0.94 |
| G2.10 | 0.31 | 0.06 | 0.69 | 0.97 | 0.93 | 0.73 | 0.94 | 0.50 | 0.68 |
| Mean | 0.25 | 0.29 | 0.69 | 0.75 | 0.84 | 0.77 | 0.71 | 0.64 | 0.73 |
| STD | 0.21 | 0.21 | 0.13 | 0.31 | 0.19 | 0.18 | 0.20 | 0.26 | 0.13 |

**Table A4** Intra-limb coordination - traditional procrustes dissimilarity indices.

| | U & F | | | U & H | | | F & H | | |
|---|---|---|---|---|---|---|---|---|---|
| | T1 | T2 | T3 | T1 | T2 | T3 | T1 | T2* | T3 |
| G1.01 | 0.42 | 0.37 | 0.12 | 0.76 | 0.57 | 0.21 | 0.56 | 0.53 | 0.10 |
| G1.02 | 0.60 | 0.79 | 0.26 | 0.62 | 0.53 | 0.20 | 0.49 | 0.73 | 0.15 |
| G1.03 | 0.65 | 0.78 | 0.35 | 0.80 | 0.63 | 0.28 | 0.73 | 0.85 | 0.35 |
| G1.04 | 0.50 | 0.32 | 0.20 | 0.80 | 0.38 | 0.28 | 0.83 | 0.47 | 0.35 |
| G1.05 | 0.70 | 0.82 | 0.33 | 0.96 | 0.72 | 0.38 | 0.85 | 0.74 | 0.36 |
| G1.06 | 0.87 | 0.53 | 0.29 | 0.93 | 0.41 | 0.45 | 0.88 | 0.55 | 0.25 |
| G1.07 | 0.70 | 0.51 | 0.62 | 0.78 | 0.65 | 0.38 | 0.77 | 0.62 | 0.64 |
| G1.08 | 0.52 | 0.78 | 0.37 | 0.87 | 0.37 | 0.44 | 0.85 | 0.74 | 0.21 |
| G1.09 | 0.52 | 0.47 | 0.47 | 0.84 | 0.63 | 0.40 | 0.69 | 0.58 | 0.46 |
| G1.10 | 0.41 | 0.27 | 0.47 | 0.56 | 0.51 | 0.32 | 0.38 | 0.55 | 0.39 |

**Table A4** (*continued*)

| | U & F | | | U & H | | | F & H | | |
|---|---|---|---|---|---|---|---|---|---|
| | T1 | T2 | T3 | T1 | T2 | T3 | T1 | T2* | T3 |
| Mean | 0.59 | 0.56 | 0.35 | 0.79 | 0.54 | 0.33 | 0.71 | 0.64 | 0.33 |
| STD | 0.14 | 0.21 | 0.15 | 0.12 | 0.12 | 0.09 | 0.17 | 0.12 | 0.16 |
| G2.01 | 1.00 | 0.96 | 0.35 | 0.96 | 0.68 | 0.28 | 1.00 | 0.93 | 0.33 |
| G2.02 | 0.67 | 0.40 | 0.16 | 0.84 | 0.63 | 0.21 | 0.85 | 0.79 | 0.26 |
| G2.03 | 0.37 | 0.40 | 0.80 | 0.69 | 0.35 | 0.63 | 0.57 | 0.57 | 0.36 |
| G2.04 | 0.65 | 0.71 | 0.41 | 0.83 | 0.57 | 0.63 | 0.84 | 0.79 | 0.54 |
| G2.05 | 0.37 | 0.92 | 0.38 | 0.72 | 0.73 | 0.44 | 0.48 | 0.91 | 0.19 |
| G2.06 | 0.68 | 0.87 | 0.68 | 0.51 | 0.59 | 0.27 | 0.76 | 0.92 | 0.50 |
| G2.07 | 0.58 | 0.83 | 0.24 | 0.89 | 0.86 | 0.26 | 0.85 | 0.83 | 0.38 |
| G2.08 | 0.58 | 0.59 | 0.82 | 0.89 | 0.94 | 0.89 | 0.85 | 0.94 | 0.90 |
| G2.09 | 0.95 | 0.57 | 0.47 | 0.87 | 0.84 | 0.38 | 0.99 | 0.79 | 0.70 |
| G2.10 | 0.69 | 0.50 | 0.31 | 0.81 | 0.28 | 0.19 | 0.87 | 0.53 | 0.30 |
| Mean | 0.65 | 0.67 | 0.46 | 0.80 | 0.65 | 0.42 | 0.81 | 0.80 | 0.45 |
| STD | 0.20 | 0.21 | 0.23 | 0.13 | 0.21 | 0.23 | 0.16 | 0.14 | 0.22 |

## Funding

This work was supported by a Dissertation Completion grant awarded by George Mason University. The APC for this article was paid by NIH: EY029715. The funders had no role in study design, data collection and analysis, decision to publish, or preparation of the manuscript.

## Grant Disclosures

The following grant information was disclosed by the authors:
George Mason University.
NIH: EY029715.

## Competing Interests

The authors declare there are no competing interests.

## Author Contributions

- Khadija F. Zaidi conceived and designed the experiments, performed the experiments, analyzed the data, prepared figures and/or tables, authored or reviewed drafts of the article, and approved the final draft.
- Qi Wei conceived and designed the experiments, authored or reviewed drafts of the article, and approved the final draft.

## Human Ethics

The following information was supplied relating to ethical approvals (i.e., approving body and any reference numbers):

This protocol was approved by the George Mason University Institutional Review Board under protocol number [2077208-1].

## Data Availability

All de-identified data is available at Zenodo: Zaidi, K. F. (2023). Age-related differences in Bilateral Synergistic Coordination of the Upper Extremity [Data set]. Zenodo. https://doi.org/10.5281/zenodo.10020020.

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
