# Peer review of "Temporal localization of upper extremity bilateral synergistic coordination using wearable accelerometers"

_PeerJ, doi:10.7717/peerj.17858_

## Round 0.1 · original submission · Major Revisions

Please, address point-by-point all reviewers' issues (especially Reviewer 2's)

Reviewer 1 ·

Basic reporting

no comment

Experimental design

・The instructions are well-detailed and clear in lines 122-153. To enhance clarity both visually and conceptually, creating a flowchart with diagrams would further improve understanding.

・The specific criteria used for selecting the three tasks have not been explicitly outlined in the document. To enhance transparency and emphasize the validity of the research design, we recommend providing a clear explanation of the criteria used for the selection of these tasks. This additional information will aid reviewers in understanding the rationale behind the task choices and contribute to the overall clarity of the study.

Validity of the findings

To further extend the applicability of the study findings, it may be worthwhile to conduct similar investigations with populations such as stroke patients or orthopedic patients. Drawing comparisons between the current participants and individuals from these cohorts could offer valuable insights for the development of comprehensive rehabilitation strategies. This comparative approach could enhance the generalizability of the study's implications and contribute to the broader understanding of motor performance across different clinical conditions.

·

Basic reporting

The article presents the issue of using movement analysis measures to assess age-related differences in movement kinematics. The issue presented in the article is interesting, providing a basis for development and practical applications, e.g. to assess the effects of rehabilitation or the ability to perform a specific job. Given the generally positive assessment of the article, there are some areas that require expansion or clarification. My main reservations concern the size of the study group and the statistical methods used.

Experimental design

- The research included two groups of 10 people each. How was the number of samples calculated, with what statistical assumptions? This issue is particularly important in view of the fact that the comparison of independent groups of data was carried out using a parametric two-way ANOVA.
- A subsection describing the statistical analysis could be expected in the Methods section. A more detailed and clearer description of the use of statistical tests was expected with underling the support that the given test provides for hypothesis.
- What test was used to check for equality of variances in data sets for two way ANOVA?
- “The Shapiro-Wilk test was performed, due to the sample size of less than 50 people, to assess the normality of age distributions in group 1 [p = 0.39] and group 2 [p = 0.077].” – there seems to be an error in this sentence. It should be expected to test mostly the distribution of the dependent data set to be analyzed.
- Perhaps, due to the small number of study participants, it would be advisable to conduct an analysis using non-parametric tests that only indicate differences between groups in relation to a given task or measures pooled over tasks?
- During the experiments, each person performed three tasks. Was the order in which the tasks were performed random? How the 1-minute break between tasks was determined (on which basis). Is 1 minute break enough to avoid fatigue? What about learning effect?

Validity of the findings

no comment

Additional comments

- The tasks differed in terms of movement phases. Doesn't that mean the differences in metrics between tasks are obvious? This issue requires discussion in order to dispel doubts.
- The discussion does not refer enough to the results of the study and the hypotheses that the study was intended to prove. One would expect the discussion to be more focused on the issues raised in the hypothesis.

---

## Round 0.2 · accepted · Accept

The authors have addressed all of the reviewers' comments.
This manuscript is ready for publication.
Congratulations on the interesting work.

Reviewer 1 ·

Basic reporting

no comment

Experimental design

no comment

Validity of the findings

no comment

·

Basic reporting

no comment

Experimental design

no comment

Validity of the findings

no comment

Additional comments

no comment